# TFE3 regulates whole-body energy metabolism in cooperation with TFEB

Nunzia Pastore[1,2], Anna Vainshtein[1,2], Tiemo J Klisch[1,2], Andrea Armani[3], Tuong Huynh[1,2], Niculin J Herz[1,2], Elena V Polishchuk[4], Marco Sandri[3,5] & Andrea Ballabio[1,2,4,6,*]

## Abstract

TFE3 and TFEB are members of the MiT family of HLH–leucine zipper transcription factors. Recent studies demonstrated that they bind overlapping sets of promoters and are post-transcriptionally regulated through a similar mechanism. However, while *Tcfeb* knock-out (KO) mice die during early embryonic development, no apparent phenotype was reported in *Tfe3* KO mice. Thus raising the need to characterize the physiological role of TFE3 and elucidate its relationship with TFEB. TFE3 deficiency resulted in altered mitochondrial morphology and function both *in vitro* and *in vivo* due to compromised mitochondrial dynamics. In addition, *Tfe3* KO mice showed significant abnormalities in energy balance and alterations in systemic glucose and lipid metabolism, resulting in enhanced diet-induced obesity and diabetes. Conversely, viral-mediated TFE3 overexpression improved the metabolic abnormalities induced by high-fat diet (HFD). Both TFEB overexpression in *Tfe3* KO mice and TFE3 overexpression in *Tcfeb* liver-specific KO mice (*Tcfeb* LiKO) rescued HFD-induced obesity, indicating that TFEB can compensate for TFE3 deficiency and *vice versa*. Analysis of *Tcfeb* LiKO/*Tfe3* double KO mice demonstrated that depletion of both TFE3 and TFEB results in additive effects with an exacerbation of the hepatic phenotype. These data indicate that TFE3 and TFEB play a cooperative, rather than redundant, role in the control of the adaptive response of whole-body metabolism to environmental cues such as diet and physical exercise.

**Keywords** lipid metabolism; mitochondrial dynamics; physical exercise; TFE3; TFEB
**Subject Category** Metabolism

## Introduction

Complex organisms have developed sophisticated and efficient mechanisms to adapt their energy metabolism to environmental cues. These mechanisms typically involve both protein post-translational modifications and protein–protein interactions for acute responses, and transcriptional regulation for more chronic sustained responses. The transcription factor EB (TFEB) plays an important role in the control of cell and organismal homeostasis (Settembre *et al*, 2013b; Napolitano & Ballabio, 2016). TFEB responds to cellular nutrient levels and regulates lysosomal biogenesis and autophagy (Sardiello *et al*, 2009; Settembre *et al*, 2011). Under nutrient-rich conditions, TFEB resides in the cytoplasm and translocates to the nucleus in response to starvation and lysosomal stress (Settembre *et al*, 2011). In the liver, fasting activates TFEB, which promotes lipophagy and lipid catabolism via induced expression of the transcriptional co-activator *Pgc1α* and nuclear receptor *Pparα* (Settembre *et al*, 2013a).

TFEB belongs to the MiT family of helix-loop-helix leucine zipper transcription factors, together with TFE3, MITF and TFEC (Steingrimsson *et al*, 2004). Recently, it has been demonstrated that TFE3 and TFEB regulate overlapping sets of genes and their overexpression has similar effects (Martina *et al*, 2014). Subcellular localization of both TFE3 and TFEB is controlled by their phosphorylation status on particular serine residues (Settembre *et al*, 2011, 2012; Roczniak-Ferguson *et al*, 2012; Martina *et al*, 2014; Medina *et al*, 2015; Li *et al*, 2016). A variety of stimuli are able to induce TFE3 and TFEB cytoplasm-to-nucleus translocation (Napolitano & Ballabio, 2016). Recent studies demonstrated that pathogen infections promote TFE3 and TFEB nuclear translocation, thereby inducing the expression of several cytokines and chemokines (Visvikis *et al*, 2014; Najibi *et al*, 2016; Pastore *et al*, 2016).

In spite of the similarities between TFE3 and TFEB, it is still unclear whether these transcription factors have similar, cooperative, complementary or redundant roles in mammalian physiology. In particular, the characterization of the physiological role of TFE3 has been hampered by the lack of an apparent phenotype in *Tfe3*

1  Jan and Dan Duncan Neurological Research Institute, Texas Children Hospital, Houston, TX, USA
2  Department of Molecular and Human Genetics, Baylor College of Medicine, Houston, TX, USA
3  Department of Biomedical Science, University of Padova, Padova, Italy
4  Telethon Institute of Genetics and Medicine (TIGEM), Pozzuoli (Naples), Italy
5  Venetian Institute of Molecular Medicine, Padova, Italy
6  Medical Genetics, Department of Medical and Translational Sciences, Federico II University, Naples, Italy
   *Corresponding author. Tel: +39 081 19230607; E-mail: ballabio@tigem.it

KO mice (Steingrimsson *et al*, 2002), whereas *Tcfeb* KO mice die during embryonic development (Steingrimsson *et al*, 1998).

In this study, by analysing mice lacking *Tfe3*, we discovered that TFE3 plays a crucial role in the regulation of lipid and glucose metabolism and is necessary for proper mitochondrial dynamics and function in both liver and muscle. Moreover, epistatic analysis revealed that TFE3 and TFEB cooperate in the metabolic adaptation to physiological stressors in multiple organs.

## Results

### *Tfe3* KO mice have an altered energy balance and glucose homeostasis

To investigate the physiological role of TFE3, we performed a thorough phenotypic and metabolic analysis of previously generated *Tfe3* KO mice (Steingrimsson *et al*, 2002). No main differences were observed in body weight between wild-type (WT) and *Tfe3* KO mice fed a chow diet (Fig 1A). However, magnetic resonance imaging (MRI) revealed a 50% increase in fat mass and a significant reduction in lean mass (Fig 1B) in *Tfe3* KO mice compared to controls. These alterations could not be attributed to differences in food intake, as food consumption was similar between the two genotypes (Fig 1C). *Tfe3* KO mice displayed enlarged fat depots and increased epididymal (eWAT) and inguinal (iWAT) fat weight with significant reduction in liver weight (Fig 1D). Consistent with increased adipose tissue weight, plasma leptin levels were elevated, while adiponectin levels demonstrated a trend towards reduction but did not reach statistical significance (Fig 1E). Comprehensive laboratory animal monitoring system (CLAMS) studies, used to measure metabolic activities, showed that during the light phase, WT and *Tfe3* KO animals had a similarly low respiration exchange ratio (RER, $V_{CO_2}/V_{O_2}$), which likely reflects the shift from carbohydrate to fatty acid consumption and the changes in activity and food intake (Fig 1F and G). *Tfe3* KO mice did not show significant differences in $O_2$ consumption (Fig EV1A and B) or $CO_2$ production (Fig EV1C and D) during the light and dark phases compared to WT controls. Notably, *Tfe3* KO mice showed reduced energy expenditure (Fig 1H and I, and Appendix Fig S1A and B) that could explain their increased adiposity.

To test whether *Tfe3* KO mice have abnormalities in glucose homeostasis, we analysed their blood glucose levels. No differences were observed among the genotypes under basal feeding conditions; however, blood glucose levels of 24-h-fasted *Tfe3* KO mice were significantly lower than those of fasted controls (Fig 2A). *Tfe3* KO mice also showed increased insulin (Fig 2B), and reduced glucagon (Fig 2C) levels in plasma. We then further examined glucose homeostasis by performing glucose tolerance test (GTT) and insulin tolerance test (ITT). While no differences were observed in ITT between the genotypes (Fig 2D), *Tfe3* KO mice showed glucose intolerance (Fig 2E), which was associated with reduced insulin secretion 15 min after glucose injection (Fig 2F). *Tfe3* KO mice also displayed altered hepatic gluconeogenesis during a PTT test (Fig 2G) and the expression of *Pepck* and *G6Pc*, key players in gluconeogenesis, was reduced in the *Tfe3* KO livers (Fig 2H), indicating insufficient gluconeogenesis as a contributing factor to the reduced fasting glucose. Periodic acid–Schiff (PAS) staining revealed a reduction in glycogen

stores in liver (Fig EV1E and F) as well as in muscle (Fig EV1G) of *Tfe3* KO mice compared to their WT littermates. During exercise, blood glucose levels and muscle glycogen stores were similarly depleted in WT and *Tfe3* KO mice (Fig 2I and J). However, whereas WT animals were able to restore their blood glucose and muscle glycogen stores following a 2-h recovery period, *Tfe3* KO animals trailed behind in their blood glucose recovery and completely failed to replete muscle glycogen stores (Fig 2I and J). Moreover, we observed a higher increase in blood lactate levels in *Tfe3* KO mice following exercise (Fig 2K), suggesting that they utilize anaerobic glycolysis as a source of energy. Failure to replete glycogen stores in muscle following exercise can occur as a result of impaired muscle glucose uptake or glycogen synthesis. Indeed, *Gsk3* expression was significantly reduced in *Tfe3* KO muscle (Fig 2L), suggesting that reduced glycogen synthesis may occur. Consistent with a role of TFE3 in glycogen homeostasis, viral-mediated *TFE3* overexpression in WT mice injected systemically using an helper-dependent adenovirus (HDAd) expressing the human *TFE3* gene under the control of a promoter mainly expressed in the liver (HDAd-PEPCK-h*TFE3*) or intramuscularly (i.m.) using an adeno-associated virus (AAV) expressing the human *TFE3* gene under the control of a ubiquitous promoter (AAV2.1-CMV-h*TFE3*) increased glycogen stores in both liver and muscle (Fig EV1H).

These data indicate that TFE3 plays a key role in regulating whole-body glucose and glycogen homeostasis.

### Abnormalities in lipid metabolism in *Tfe3* KO mice

Since *Tfe3* KO mice show an increased adiposity (Fig 1B and D), we evaluated whether lipid homeostasis was also altered. We investigated the ability of *Tfe3* KO mice to utilize fat in response to starvation. Under basal feeding conditions, livers from *Tfe3* KO mice displayed normal morphology and gross appearance with no apparent histological abnormalities (Figs 1D and EV2A). Conversely, following 24-h starvation, *Tfe3* KO mice exhibited marked hepatic steatosis as determined by gross inspection, H&E and Oil Red O staining (Fig EV2A and B). Electron microscopy (EM) analysis (Fig EV2C) and the direct quantification of total lipid content in the liver (Fig EV2D) confirmed an increased accumulation of lipids in *Tfe3* KO livers upon fasting. Moreover, several metabolic parameters (ALT, AST, CK, LDH) were significantly increased in *Tfe3* KO mice in response to starvation (Appendix Table S1) suggesting liver stress. A reduced expression of genes involved in lipid metabolism (e.g. *Cd36*, *Cyp7a1*, *Fgf21*, *Cpt1α*, *Pgc1α*, *ApoA4*, *Cyp17a1*, *Cyp4a10*, *Cyp4a14*) was detected in 24-h-fasted *Tfe3* KO livers and primary hepatocytes as compared to fasted WT (Fig EV2E and F). Moreover, a reduction in the expression of genes involved in lipogenesis (e.g. *Fasn* and *Srebp1c*) was detected in *Tfe3* KO mice in the fed state (Fig EV2E). As expected, opposite effects were observed in WT mice systemically injected with the HDAd-PEPCK-h*TFE3*, which showed increased expression of genes involved in lipid metabolism in the liver when compared to controls (Fig EV2G).

To further evaluate lipid homeostasis, we fed *Tfe3* KO mice and controls with a high-fat diet (HFD) for 18 weeks. HFD-fed *Tfe3* KO mice gained significantly more weight (Fig 3A and B) and showed a pale liver and increased adipose mass (Fig 3A) compared to WT mice. Indeed, MRI analysis revealed a further increase in fat mass with no reduction in lean mass in *Tfe3* KO mice compared to WT

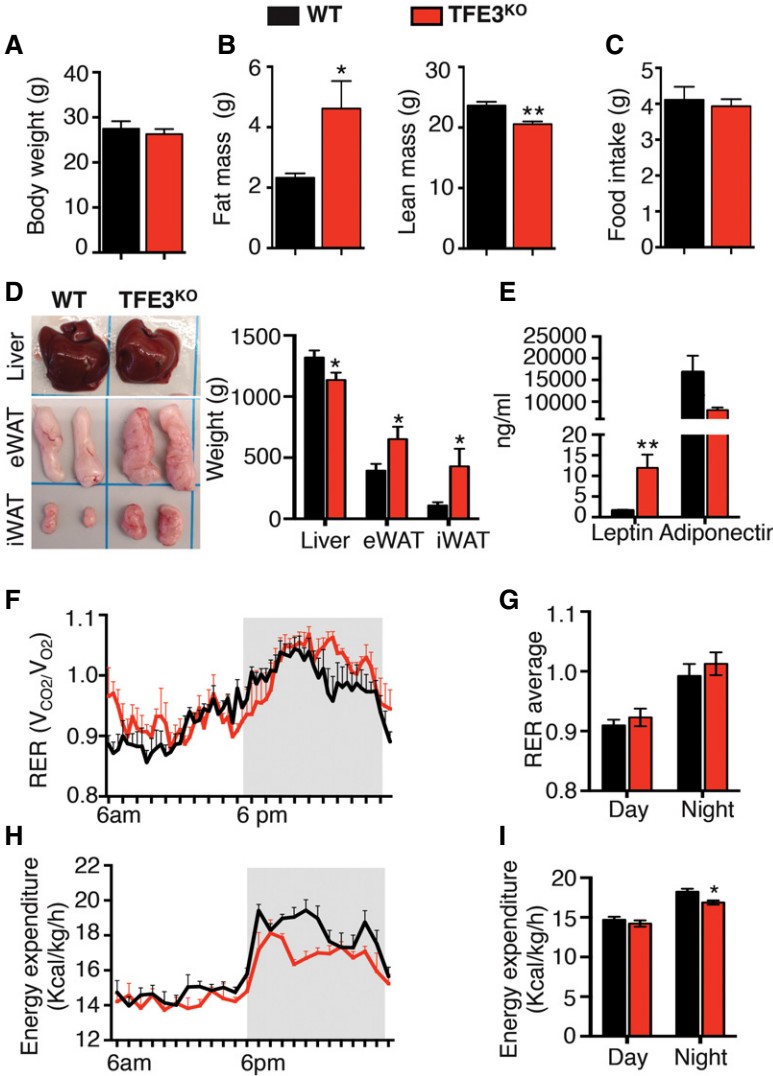

**Figure 1. TFE3 regulates energy homeostasis.**

A–D  Body weight (A) (*n* = 5 per group), fat and lean mass (B) (*n* = 5 per group), food intake (C) (*n* = 5 per group) and gross appearance and tissue weight (D) (*n* = 15 per group) from 2-month-old mice fed a chow diet with the indicated genotypes. Data are presented as mean ± SEM. Student's two-tailed *t*-test: fat mass *P* = 0.0257; lean mass **P* = 0.0066; liver weight *P* = 0.0344; eWAT weight *P* = 0.0398; iWAT weight *P* = 0.0415.

E  Serum leptin and adiponectin levels in *Tfe3* KO mice compared to control mice (*n* = 3 per group). Data are presented as mean ± SEM. Student's two-tailed *t*-test: **P* = 0.0045.

F  Respiratory exchange ratio (RER; $V_{CO2}/V_{O2}$) in WT (black line; *n* = 5) and *Tfe3* KO mice (red line; *n* = 4). Grey areas indicate dark periods (6 PM to 6 AM). Data are presented as mean ± SEM.

G  Bar graph represents average RER values during day and night (*n* = 5 WT and *n* = 4 *Tfe3* KO). Data are presented as mean ± SEM.

H  Energy expenditure (EE) normalized for body weight in WT (black line; *n* = 5) and *Tfe3* KO mice (red line; *n* = 4). Grey areas indicate dark periods (6 PM to 6 AM). Data are presented as mean ± SEM.

I  Bar graph represents average EE values during day and night (*n* = 5 WT and *n* = 4 *Tfe3* KO). Data are presented as mean ± SEM. Student's two-tailed *t*-test: *P* = 0.0307.

(Fig 3C). This was not due to increased food intake, which did not significantly differ between the genotypes (Fig EV3A). *Tfe3* KO mice showed hyperleptinaemia, hypoadiponectinaemia, hyperinsulinaemia and hypercholesterolaemia compared to controls (Fig EV3B). Although FFA levels showed a trend for increase, they were not significantly different (Fig EV3B). Indirect calorimetry showed a significant increase in RER ($V_{CO2}/V_{O2}$) values indicating a preferential utilization of carbohydrates and reduced lipid catabolism in *Tfe3* KO mice fed with a HFD (Fig 3D and E), which explains the obesity-prone phenotype.

They also showed increased $O_2$ consumption (Fig EV3C and D) and $CO_2$ production (Fig EV3E and F), but reduced energy expenditure per body weight (Fig 3F and G, and Appendix Fig S2A and B), which accounts for the impaired energy balance in *Tfe3* KO mice.

Excessive adiposity and obesity are linked to insulin resistance (Kahn & Flier, 2000). Consistent with these findings, *Tfe3* KO mice showed enhanced glucose intolerance and insensitivity, as well as insulin resistance, as indicated by glucose and insulin levels during GTT (Fig EV3G and H) and ITT (Fig EV3I) after 1 month of HFD.

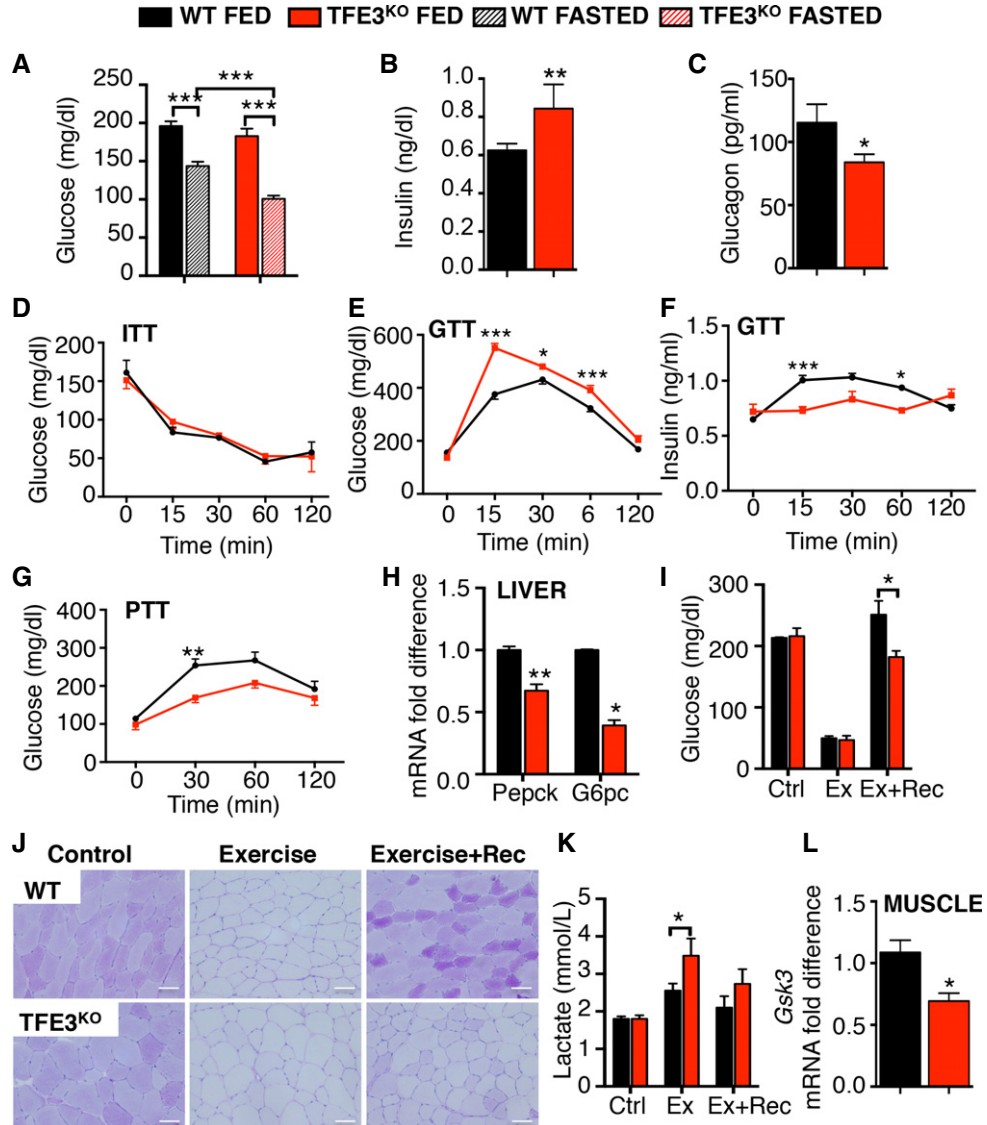

**Figure 2. TFE3 regulates glucose homeostasis.**

A   Glucose levels in fed and after 24 h fasting in 2-month-old WT and *Tfe3* KO mice (*n* = 12 per group). Data are presented as mean ± SEM. Student's two-tailed *t*-test: ****P* < 0.001.

B, C   Plasma insulin (B) (*n* = 5 WT and *n* = 3 *Tfe3* KO) and glucagon (C) (*n* = 12 per group) levels in WT and *Tfe3* KO mice. Data are presented as mean ± SEM. Student's two-tailed *t*-test: ***P* = 0.0090; **P* = 0.0404.

D–F   Insulin and glucose tolerance tests in WT and *Tfe3* KO mice fed a chow diet. Glucose levels (D) at the indicated time points after insulin challenge (*n* = 3 per group). Glucose (E) (*n* = 20 per group) and serum insulin (F) (*n* = 5 per group) levels at the indicated time points after glucose challenge. Data are presented as mean ± SEM. ANOVA test followed by *post hoc* Bonferroni test: GTT ****P* = 0.0003 15 min; **P* = 0.0241 30 min; ****P* = 0.0007 60 min; insulin during GTT ****P* = 0.0005; **P* = 0.0145.

G   Pyruvate tolerance test in WT (black line; *n* = 5) and *Tfe3* KO mice (red line; *n* = 4) fed a chow diet. Data are presented as mean ± SEM. ANOVA test followed by *post hoc* Bonferroni test: ***P* = 0.0053.

H   Expression analysis of genes involved in gluconeogenesis in the liver (*n* = 5 per group). Data are presented as mean ± SEM. Student's two-tailed *t*-test: ***P* = 0.0098; **P* = 0.0216.

I–K   Blood glucose (I) (*n* = 4 per group), muscle glycogen content (J) (scale bars 50 μm) and blood lactate levels (K) (*n* = 6 Ctrl and Ex; *n* = 3 Ex+Rec) from mice with the indicated genotypes subjected to an exercise (Ex) and exercise plus 2-h recovery protocol (Ex+Rec). Data are presented as mean ± SEM. Student's two-tailed *t*-test: blood glucose **P* = 0.049; blood lactate **P* = 0.027.

L   Expression analysis of *Gsk3* in muscle of WT and *Tfe3* KO mice fed a chow diet (*n* = 4 per group). Data are presented as mean ± SEM. Student's two-tailed *t*-test: **P* = 0.0149.

Muscle glucose uptake was also impaired in HFD-fed *Tfe3* KO mice (Fig EV3J). In line with the severe obesity, *in vivo*-stimulated lipolysis was impaired in *Tfe3* KO mice (Fig EV3K). Histological analysis

performed after 18 weeks of HFD revealed a marked vacuolar degeneration and fat accumulation in the liver of *Tfe3* KO mice (Fig 3H and I), as well as enlarged lipid droplets in both eWAT and

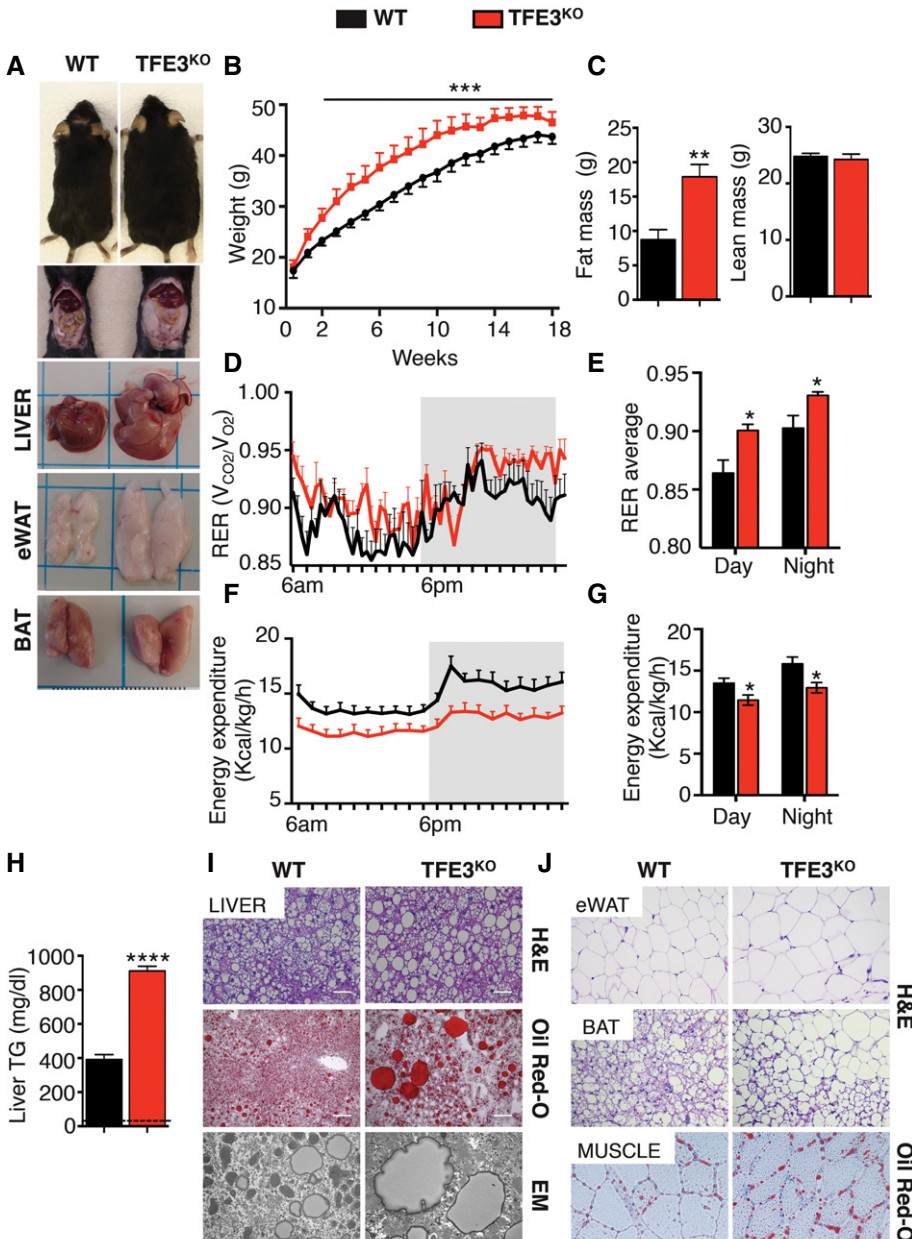

**Figure 3.  TFE3 regulates lipid catabolism.**

A   Body size and tissues' gross appearance from WT and *Tfe3* KO mice after 1 month of high-fat diet (HFD).

B   Body weight gained over time in control and *Tfe3* KO mice fed a HFD for 18 weeks (*n* = 6 per group). Data are presented as mean ± SEM. ANOVA test followed by *post hoc* Bonferroni test: ***P < 0.001.

C   Visceral fat and lean mass from mice with the indicated genotypes after 1 month on HFD (*n* = 5 per group). Data are presented as mean ± SEM. Student's two-tailed *t*-test: **P = 0.0041.

D   Respiratory exchange ratio (RER; $V_{CO_2}/V_{O_2}$) in WT (black line) and *Tfe3* KO (red line) mice after 1 month of HFD (*n* = 5 per group). Grey areas indicate dark periods (6 PM to 6 AM). Data are presented as mean ± SEM.

E   Bar graph represents RER average values during day and night (*n* = 5 per group). Data are presented as mean ± SEM. Student's two-tailed *t*-test: day *P = 0.0149; night *P = 0.0270.

F   Energy expenditure (EE) normalized for body weight in WT (black line) and *Tfe3* KO mice (red line) after 1 month of HFD (*n* = 5 per group). Grey areas indicate dark periods (6 PM to 6 AM). Data are presented as mean ± SEM.

G   Bar graph represents average EE values during day and night (*n* = 5 per group). Data are presented as mean ± SEM. Student's two-tailed *t*-test: day *P = 0.0485; night *P = 0.0227.

H   Liver triglycerides (TG) levels in WT and *Tfe3* KO mice after 18 weeks of HFD (*n* = 5 per group). Dashed line indicates WT fed a chow diet. Data are presented as mean ± SEM. Student's two-tailed *t*-test: ****P < 0.0001.

I, J   H&E, Oil Red O staining and EM images of liver sections (I) and H&E staining of eWAT, BAT and muscle sections (J) from *Tfe3* KO and control mice fed a HFD for 18 weeks (scale bars liver, eWAT, BAT: 50 μm; scale bars muscle: 20 μm).

    

brown adipose tissue (BAT; Fig 3J). Importantly, *Tfe3* KO mice showed down-regulation of genes involved in thermogenesis, including *Ucp1* and *Ucp3*, and oxidative genes such as *Pparα* in the BAT (Appendix Fig S3A and B), which account, at least partially, for their reduced energy expenditure and diet-induced obesity. A higher content of lipid droplets was also observed in skeletal muscle after HFD (Fig 3J).

Thus, the absence of TFE3 in mice results in enhanced diet-induced obesity and diabetes, indicating that TFE3 plays an important role in lipid metabolism, metabolic homeostasis and in the adaptive response to dietary lipids.

## TFE3 controls the response to physical exercise

Exercise elicits several beneficial effects by acting on mitochondrial content/function, fatty acid oxidation and glucose homeostasis (Holloszy & Coyle, 1984; Holloszy *et al*, 1998; Hawley, 2002). Physical activity is also important to counteract disease progression in diabetes, obesity and metabolic syndrome. *Tfe3* KO mice performed significantly worse compared to their WT littermates in an exhaustive endurance test (Fig 4A), indicating impaired endurance capacity. In order to test whether TFE3 is activated by exercise, we electroporated a CMV-TFE3-GFP plasmid in the tibialis anterior (TA) muscle of WT mice and performed an acute bout of exercise 8 days later. As evident by increased nuclear GFP signal in muscle and subcellular fractionation analysis in liver, TFE3, as well as TFEB, translocate to the nucleus in response to exercise (Fig 4B–D). We then subjected WT and *Tfe3* KO mice fed with a HFD to 8 weeks of exercise training. While WT mice greatly benefited from chronic exercise training, as evident by the reduced body (Fig 4E and F), liver (Fig 4G) and WAT weight gain (Fig 4H) and improved glucose tolerance (Fig 4I), *Tfe3* KO mice did not show the same amelioration in adiposity (Fig 4F and H) and glucose tolerance (Fig 4I) and displayed a reduced improvement in body weight (Fig 4F) and endurance capacity compared to the WT mice (Fig 4J).

These findings implicate TFE3 in mediating exercise-induced metabolic adaptations.

## TFE3 regulates mitochondrial dynamics and function

Electron microscopy (EM) analysis of *Tfe3* KO liver and muscle revealed an unexpected and pronounced accumulation of morphologically abnormal mitochondria (Fig 5A and B). In liver, there was no difference detected in mitochondrial number, but a significant increase in size was observed (Fig 5C), while mitochondrial number and size were both increased in muscle (Fig 5D). These data suggest that an alteration in mitochondrial dynamics may be at play.

During starvation, mitochondria elongate in WT primary hepatocytes as demonstrated by a network of highly interconnected organelles (Fig 5E), as previously reported (Gomes *et al*, 2011). In contrast, *Tfe3* KO hepatocytes show round, bigger mitochondria, which remain enlarged but punctate regardless of nutritional status (Fig 5E). Indeed, the expression of genes involved in mitochondrial dynamics (e.g. *Fis1, Drp1* and *Mfn1*) was reduced in *Tfe3* KO liver (Fig 5F). In contrast, overexpression of TFE3 in liver induced the expression of genes involved in mitochondrial dynamics (*Fis1, Drp1, Opa1, Mfn1* and *Mfn2*; Fig 5F). To understand whether TFE3

directly regulates mitochondrial gene expression, we analysed published chromatin immunoprecipitation sequencing (ChIP-seq) data of TFE3 in mouse ES cells (Betschinger *et al*, 2013). Gene ontology analysis identified 136 mitochondria-related genes targeted by TFE3. Analysis of the mitochondrial biological processes revealed that the fission pathway was highly enriched (Appendix Table S2). Indeed, *Fis1*, a mitochondrial fission receptor, was among the most highly enriched genes. Promoter analysis revealed two E-boxes in the promoter of *Fis1* close to the TSS. ChIP-qPCR experiments in TFE3-overexpressing and KO livers confirmed that TFE3 binds directly to the *Fis1* promoter, thus directly regulating its expression (Fig 5G).

We then tested whether the abnormalities in mitochondrial dynamics observed in *Tfe3* KO mice were associated with defective mitochondrial biogenesis and function. We investigated the integrity of the electron transport chain (ETC), which is composed of sequentially acting electron carriers consisting of integral membrane proteins with prosthetic groups. Protein levels of complexes IV and V, as indicated by cytochrome *c* oxidase subunit 1 (mt-Co1) and ATP synthase, $H^+$ transporting, mitochondrial F1 complex, alpha subunit 1 (Atp5a1), respectively, were not significantly changed in liver (Fig 6A and B) of *Tfe3* KO mice. In contrast, complexes I, II and III, NADH:ubiquinone oxidoreductase subunit B8 (Ndufb8), succinate dehydrogenase complex, subunit B (Sdhb) and ubiquinol-cytochrome *c* reductase core protein 2 (Uqcrc2), were significantly down-regulated (Fig 6A and B) in *Tfe3* KO livers. Consistently, TFE3 overexpression in the liver results in significant increase in complex I, II and III protein levels (Fig 6A and B).

Mitochondria are the main source of reactive oxygen species (ROS) in the cell and reduced complex I activity promotes mitochondrial ROS production (Verkaart *et al*, 2007). In line with the observed reduction in complex I activity, oxidative stress, as revealed by protein carbonylation, was significantly higher in *Tfe3* KO livers compared to controls in line with their reduced complex I activity (Appendix Fig S4A). To determine whether mitochondria were responsible for the oxidative stress detected in *Tfe3* KO mice, we transfected mouse embryonic fibroblasts from WT and *Tfe3* KO mice with a ratio metric mitochondrial targeted ROS sensor. Confocal microscopy analysis demonstrated increased ROS levels in *Tfe3* KO MEFs compared to controls (Appendix Fig S4B).

To evaluate mitochondrial function, we assessed mitochondrial membrane potential with TMRM in TFE3-depleted primary hepatocytes, HeLa and MEFs. Addition of oligomycin, an ATP-synthase inhibitor, caused a more robust reduction in membrane potential in *Tfe3*-depleted cells compared to control cells, while the protonophore carbonylcyanide-p-trifluoromethoxyphenyl hydrazine (FCCP), which was used as a positive control, completely depolarized both WT and *Tfe3* KO cells (Fig 6C and D, and Appendix Fig S4C and D), indicating that mitochondria of *Tfe3*-depleted cells are compromised and require the reverse activity of ATP synthase to maintain their membrane potential. We then measured the respiratory capacity of primary hepatocytes lacking *Tfe3* using the seahorse system. *Tfe3* KO hepatocytes showed a significant reduction in oxygen consumption rate (OCR) and markedly reduced spare respiratory capacity (SRC; Fig 6E). Moreover, basal oxygen consumption rate (OCR) and the uncoupled respiration were reduced, while coupling efficiency was unchanged (Fig 6F) in *Tfe3* KO hepatocytes compared to controls, indicating a reduction in oxidative

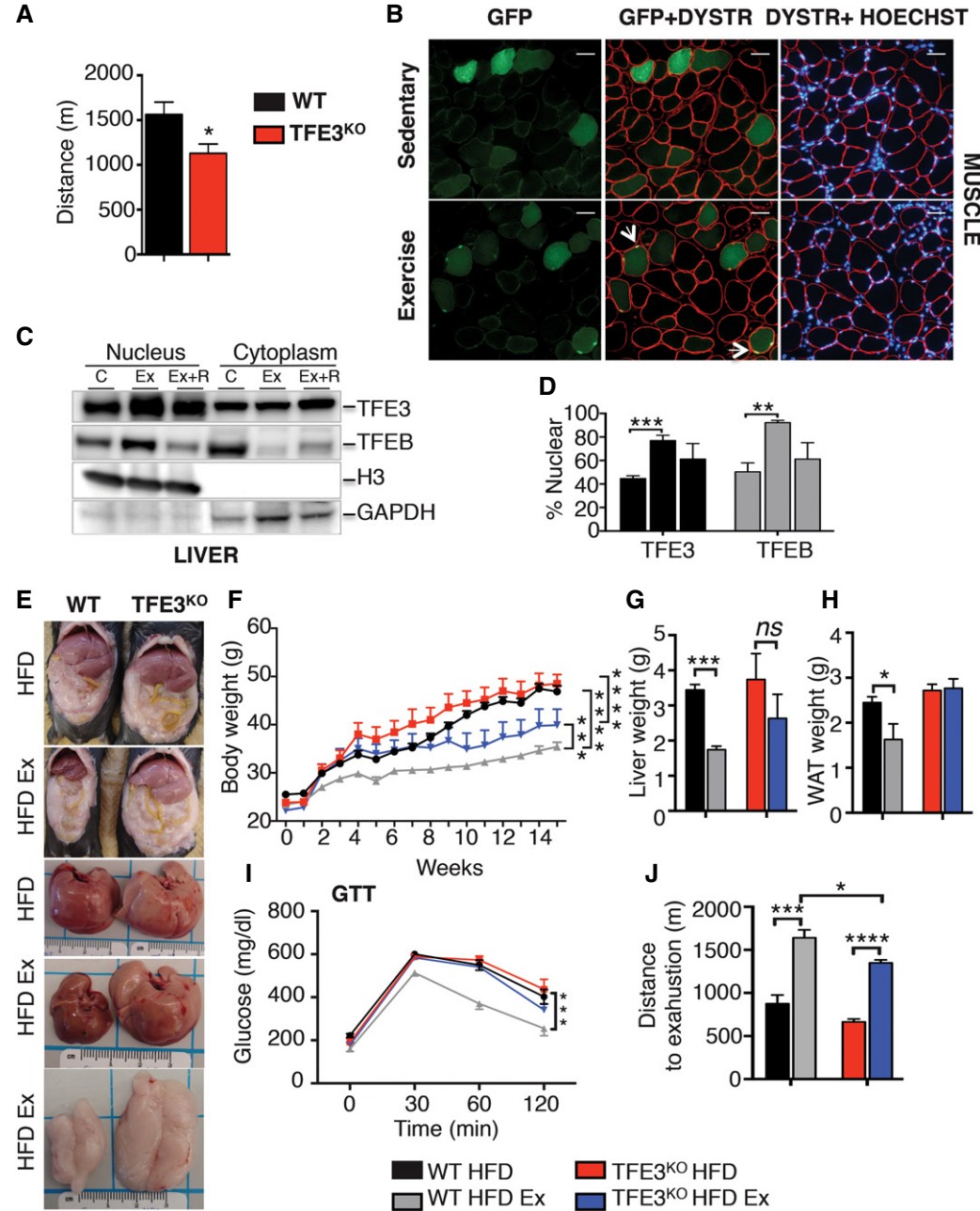

**Figure 4.  TFE3 is required for exercise-induced metabolic benefits.**

A       Distance run to exhaustion by WT and *Tfe3* KO mice fed a chow diet on an exhaustive endurance exercise test (*n* = 6 per group). Data are presented as mean ± SEM. Student's two-tailed *t*-test: *P = 0.0333.
B       Exercise-mediated nuclear translocation of TFE3 in tibialis anterior (TA) muscle from WT mice electroporated with a TFE3-GFP plasmid (scale bars: 50 μm). Arrows indicate GFP nuclear localization upon exercise.
C, D    Representative immunoblots of enriched nuclear and cytosolic cellular subfraction from livers of WT exercised (Ex) and exercised mice that were allowed to recover for 2 h (Ex+R) (C) and relative quantification (D) (*n* = 4). Data are presented as mean ± SEM. Student's two-tailed *t*-test: ***P = 0.0007; **P = 0.0017.
E, F    Gross appearance (E) and body weight (F) of WT and *Tfe3* KO mice fed a HFD with and without chronic exercise training (Ex) (*n* = 6 per group) at the end of the experiment. Data are presented as mean ± SEM. ANOVA test followed by *post hoc* Bonferroni test: ***P < 0.001; ****P < 0.0001.
G, H    Liver (G) and WAT (H) weight of WT (*n* = 6) and *Tfe3* KO (*n* = 4) mice fed a HFD with and without exercise training (Ex). Data are presented as mean ± SEM. Student's two-tailed *t*-test: ***P < 0.001; *P = 0.0112.
I       Glucose tolerance test in WT and *Tfe3* KO mice fed a HFD with and without chronic exercise training (*n* = 3 per group). Data are presented as mean ± SEM. ANOVA test followed by *post hoc* Bonferroni test: ***P < 0.001.
J       Distance run to exhaustion by exercise-trained and sedentary WT and *Tfe3* KO mice fed a HFD on an endurance exercise test (*n* = 5 per group). Data are presented as mean ± SEM. Student's two-tailed *t*-test: ***P = 0.0008; ****P < 0.0001; *P = 0.0250.

Source data are available online for this figure.

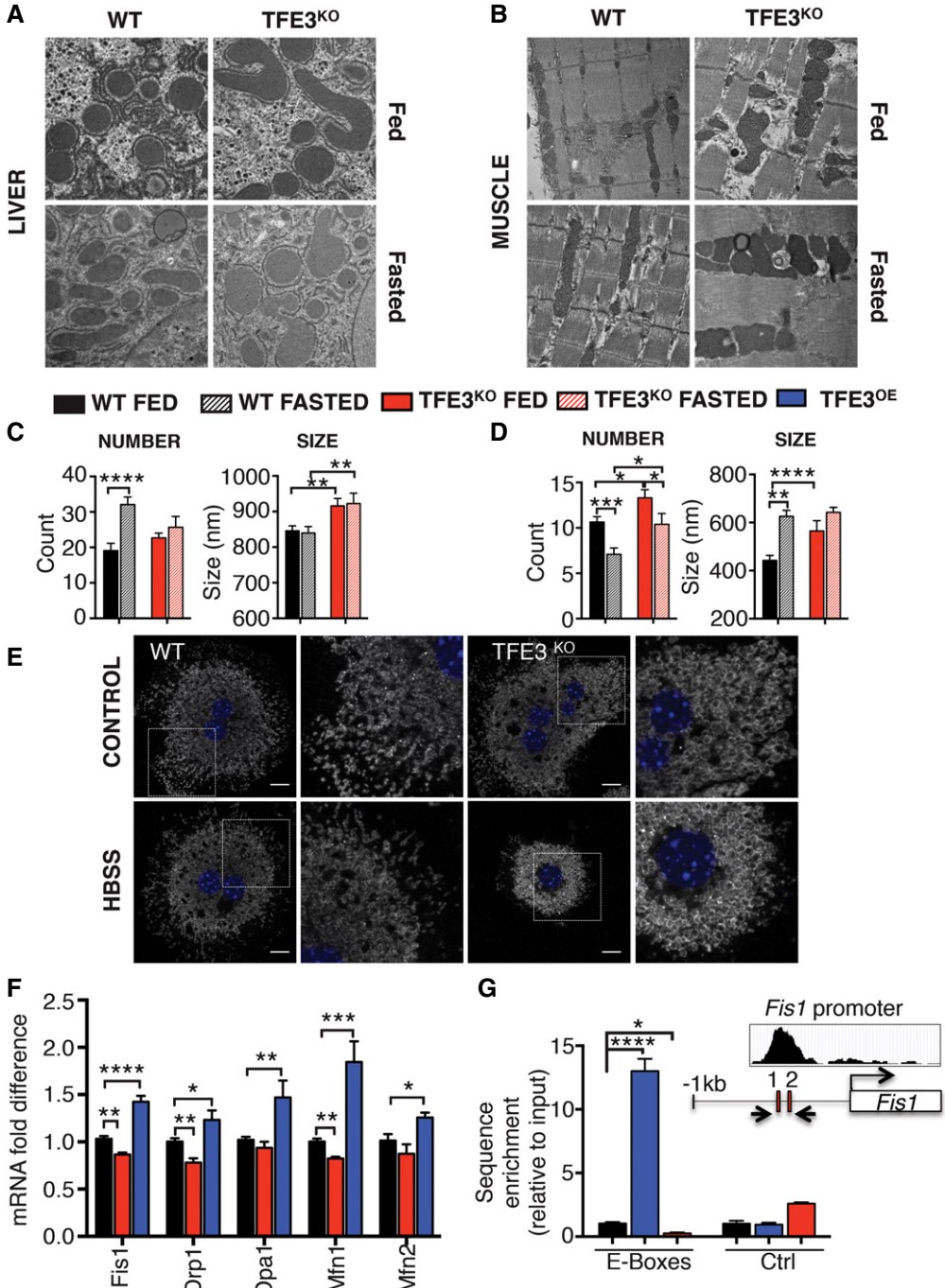

**Figure 5. TFE3 controls mitochondrial dynamics.**

A, B    Electron microscopy analysis of liver (A) and skeletal muscle (B) from WT and *Tfe3* KO mice fed a chow diet or 24-h-fasted mice showing mitochondrial morphology.

C, D    Morphometrical analysis of liver (C) and skeletal muscle (D) of fed and 24-h-fasted *Tfe3* KO mice revealed higher number and increased size of mitochondria compared to control mice. Quantification has been performed as reported in the Materials and Methods section. Data are presented as mean ± SEM. Student's two-tailed *t*-test: liver ***$P$ = 0.001; **$P$ = 0.01 WT fed versus *Tfe3* KO fed; **$P$ = 0.01 WT fasted versus *Tfe3* KO fasted. Muscle number ***$P$ = 0.0005; *Tfe3* KO fed versus *Tfe3* KO fasted *$P$ = 0.05; WT fed versus *Tfe3* KO fed *$P$ = 0.0190; WT fasted versus *Tfe3* KO fasted *$P$ = 0.0175. Muscle size ****$P$ = 1.45E-05; **$P$ = 0.0066.

E    Immunofluorescence analysis for TOM20 in primary hepatocytes isolated from WT and *Tfe3* KO mice in control media or HBSS for 1 h (scale bars: 10 μm).

F    Expression analysis of the genes involved in mitochondrial dynamics in liver from WT, *Tfe3* KO and WT mice injected with an HDAd-PEPCK-h*TFE3* virus resulting in liver-specific TFE3 expression (n = 4 per group). Data are presented as mean ± SEM. Student's two-tailed *t*-test: *Fis1* **$P$ = 0.0019, ****$P$ < 0.0001; *Drp1* **$P$ = 0.0071, *$P$ = 0.0261; *Opa1* **$P$ = 0.0029; *Mfn1* **$P$ = 0.0076, ***$P$ = 0.0006; *Mfn2* *$P$ = 0.0393.

G    ChIP analysis of *Fis1* promoter in liver extracts from WT, *Tfe3* KO or TFE3-overexpressing mice and relative controls. Data are presented as mean ± SEM of two independent experiments. Student's two-tailed *t*-test: ****$P$ < 0.0001; *$P$ = 0.0158.

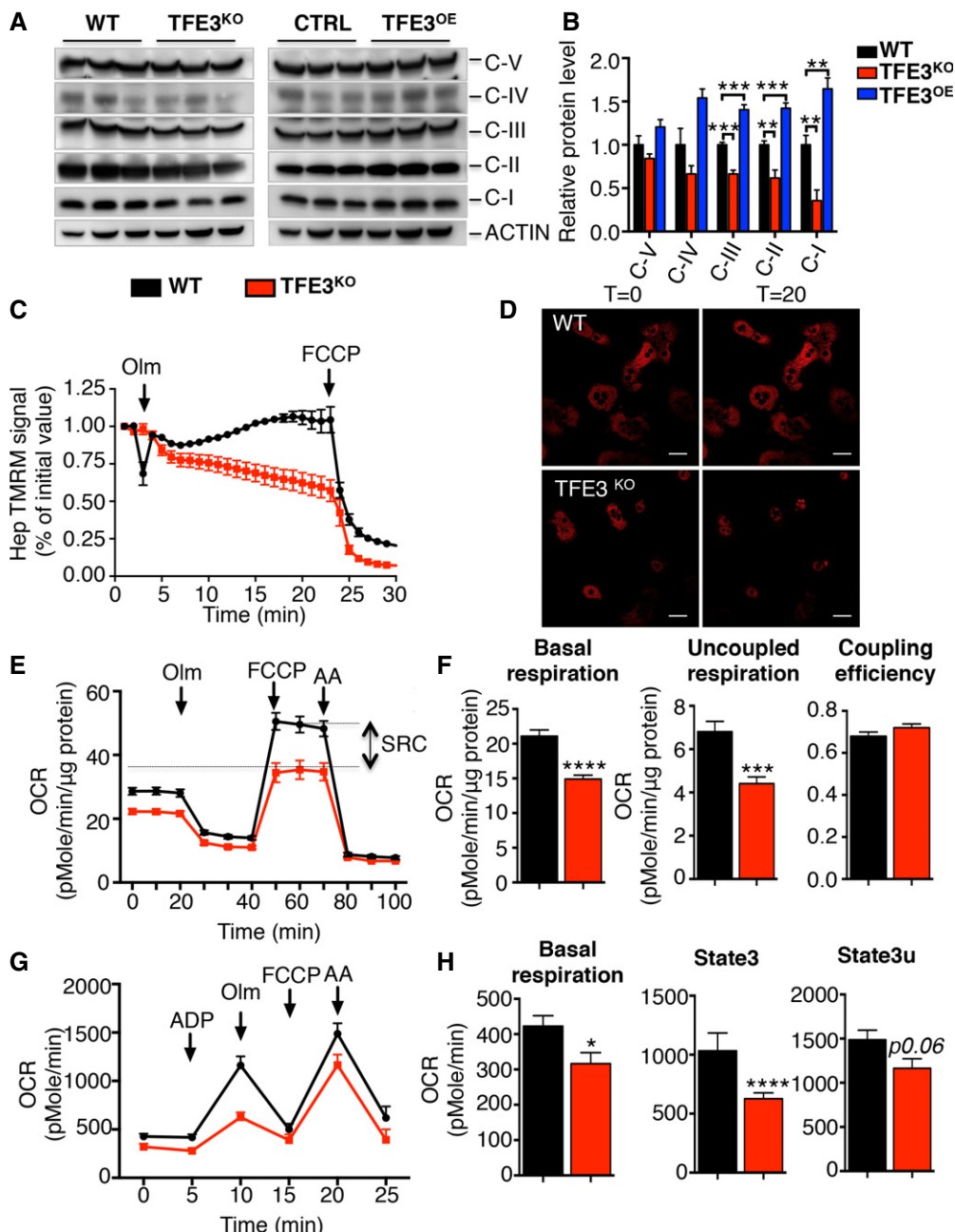

**Figure 6.  TFE3 regulates mitochondrial function.**

A, B   OXPHOS protein levels in liver from WT, *Tfe3* KO and TFE3-overexpressing mice fed a chow diet (A) and relative protein quantification (B) (*n* = 3 per group). Data are presented as mean ± SEM. Student's two-tailed *t*-test: CIII *Tfe3* KO ***P = 0.0003, *Tfe3*OE ***P = 0.0002; CII *Tfe3* KO **P = 0.0036, *Tfe3*OE ***P = 0.0004; CI *Tfe3* KO **P = 0.0088, *Tfe3*OE **P = 0.0049.

C   Mitochondrial membrane potential in primary hepatocytes isolated from WT and *Tfe3* KO mice (*n* = 5 WT and *n* = 8 *Tfe3* KO). TMRM signal was quantified as percentage of initial value. Where indicated, oligomycin (olm) or protonophore carbonylcyanide-p-trifluoromethoxyphenyl hydrazone (FCCP) was added. Data are presented as mean ± SEM.

D   Representative images of TMRM signal at the indicated time points after olm addition (scale bars: 100 μm).

E   Bioenergetic assay in hepatocytes from control and *Tfe3* KO mice (*n* = 4). Where indicated, olm, FCCP or antimycin A (AA) was added. Data were normalized for protein content. Data are presented as mean ± SEM.

F   The basal OCR, uncoupled respiration (proton leak) and uncoupling efficiency (ATP turnover/basal OCR) calculated based on data in (E). Data are presented as mean ± SEM. Student's two-tailed *t*-test: ****P < 0.0001; ***P = 0.0003.

G   Coupling assay in mitochondria isolated from WT and *Tfe3* KO livers (*n* = 4). Where indicated, ADP, olm, FCCP or antimycin A (AA) was added. Data are presented as mean ± SEM.

H   OCR basal, at state 3 and state 3u determined based on data in (G). Data are presented as mean ± SEM. Student's two-tailed *t*-test: *P = 0.0309; ****P < 0.0001.

Source data are available online for this figure.

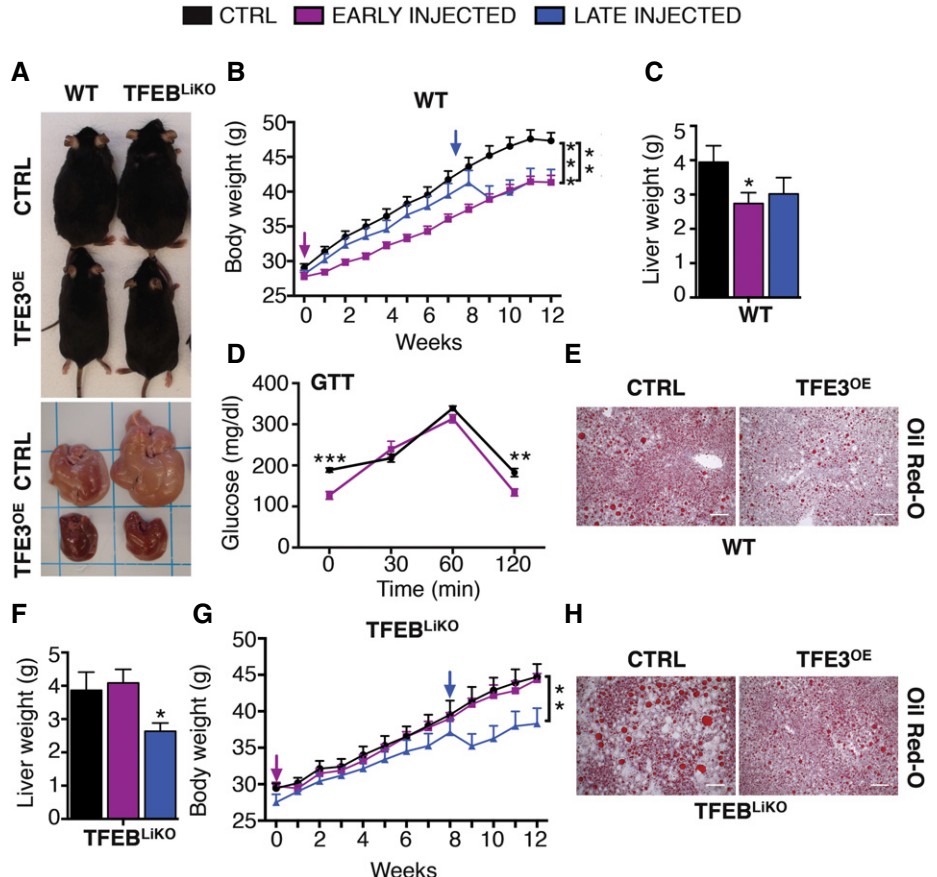

**Figure 7. TFE3 prevents diet-induced obesity and metabolic syndrome.**

A   Gross appearance of WT and *Tcfeb* LiKO mice and livers that were either uninjected (control) or injected systemically with a HDAd-PEPCK-h*TFE3* 12 weeks after HFD.
B   Body weight gain in WT mice that were injected with the HDAd-PEPCK-h*TFE3* prior to HFD administration (early) or 8 weeks into HFD (late) (*n* = 14 per group). Arrows indicate the time of injection. Data are presented as mean ± SEM. ANOVA test followed by *post hoc* Bonferroni test: **$P < 0.01$; ***$P < 0.0001$.
C   Liver weight of control (*n* = 9), early (*n* = 10)- and late (*n* = 6)-injected WT mice. Data are presented as mean ± SEM. Student's two-tailed *t*-test: *$P = 0.0473$.
D   Glucose tolerance test in control (*n* = 5) and HDAd-PEPCK-h*TFE3*-injected mice (*n* = 4) 1 month after the injection. Data are presented as mean ± SEM. ANOVA test followed by *post hoc* Bonferroni test: ***$P = 0.0007$; **$P = 0.0087$.
E   Oil Red O staining of liver sections from WT and late-injected WT mice following 12 weeks of HFD treatment (Scale bars: 100 μm).
F   Liver weight of *Tcfeb* LiKO mice that were injected with the HDAd-PEPCK-h*TFE3* prior to HFD administration (early) (*n* = 10) or 8 weeks into HFD (late) (*n* = 6) and controls (*n* = 6). Data are presented as mean ± SEM. Student's two-tailed *t*-test: *$P = 0.0184$.
G   Body weight of *Tcfeb* LiKO control (*n* = 9), early (*n* = 5)- and late-injected (*n* = 8) mice. Arrows indicate the time of injection. Data are presented as mean ± SEM. ANOVA test followed by *post hoc* Bonferroni test: **$P < 0.01$.
H   Oil Red O staining of liver sections from control *Tcfeb* LiKO and late-injected mice following 12 weeks of HFD treatment. Scale bars: 100 μm.

phosphorylation. In addition, mitochondria isolated from *Tfe3* KO livers showed a reduction in OCR in all stages (basal, state 3 and state 3u) compared to those from WT livers (Fig 6G and H). Collectively, these results implicate TFE3 in modulating mitochondrial function.

**TFE3 cooperates with TFEB in the control of energy metabolism**

To investigate whether TFE3 is sufficient to mitigate HFD-induced obesity and diabetes, we overexpressed TFE3 in the liver of WT mice by intravenous injection of the virus expressing the human *TFE3* under the control of a promoter mainly expressed in the liver (HDAd-PEPCK-*TFE3*) and fed them with a HFD for 12 weeks. HFD feeding resulted in increased body weight in control mice (Fig 7A and B), while both early (HDAd-PEPCK-*TFE3* injection prior to HFD

administration) and late (HDAd-PEPCK-*TFE3* injection 8 weeks into HFD) TFE3 overexpression in the liver resulted in reduced body and liver weights (Fig 7A–C) compared to control mice. GTT analysis following 4 weeks of HFD showed reduced blood glucose and improved glucose tolerance in *TFE3*-injected mice (Fig 7D). Moreover, mice overexpressing TFE3 exhibited a reduced number of lipid droplets and reduced hepatosteatosis as evident from liver sections stained with Oil Red O (Fig 7E). Furthermore, several metabolic parameters (ALT, AST, LDH) were reduced, indicating amelioration of liver stress (Appendix Table S3).

We previously demonstrated that *Tcfeb* liver-specific KO mice have abnormalities in lipid metabolism, which are significantly enhanced by HFD (Settembre *et al*, 2013a). To test whether TFE3 is sufficient to rescue the phenotype of *Tcfeb*-depleted mice, we systemically injected *Tcfeb* liver-specific KO (*Tcfeb* LiKO) mice

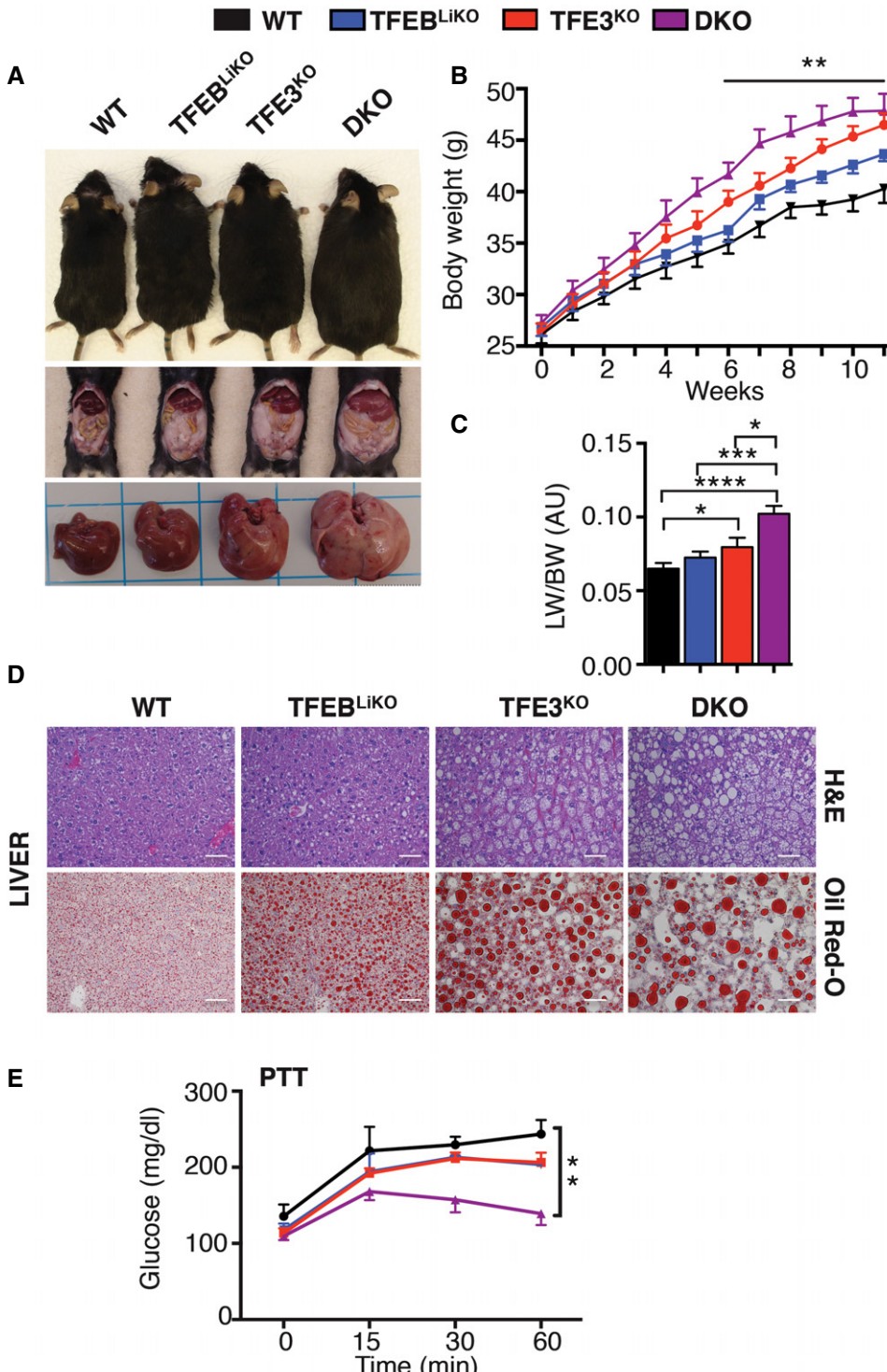

**Figure 8. TFE3 and TFEB cooperate in the control of energy metabolism.**

A   Gross appearance of WT, *Tcfeb* LiKO, *Tfe3* KO and *Tcfeb* LIKO; *Tfe3* KO (DKO) mice, liver and WAT after 12 weeks of HFD.

B   Body weight from mice of the indicated genotypes (*n* = 10 per group). Data are presented as mean ± SEM. ANOVA test followed by *post hoc* Bonferroni test: **$P < 0.01$.

C   Liver weight from mice of the indicated genotypes (*n* = 12 per group). Data are presented as mean ± SEM. Student's two-tailed *t*-test: WT versus *Tfe3* KO *$P = 0.05$; WT versus DKO ****$P < 0.0001$; *Tcfeb* LIKO versus DKO ***$P = 0.0005$; *Tfe3* KO versus DKO *$P = 0.0138$.

D   H&E (upper panel) and Oil Red O staining (lower panel) of liver sections of the indicated genotypes 12 weeks after the HFD (scale bars: 50 μm).

E   Pyruvate tolerance test (PTT) of mice at the indicated genotypes following 1 month of HFD (*n* = 3 per group). Data are presented as mean ± SEM. ANOVA test followed by *post hoc* Bonferroni test: **$P < 0.01$.

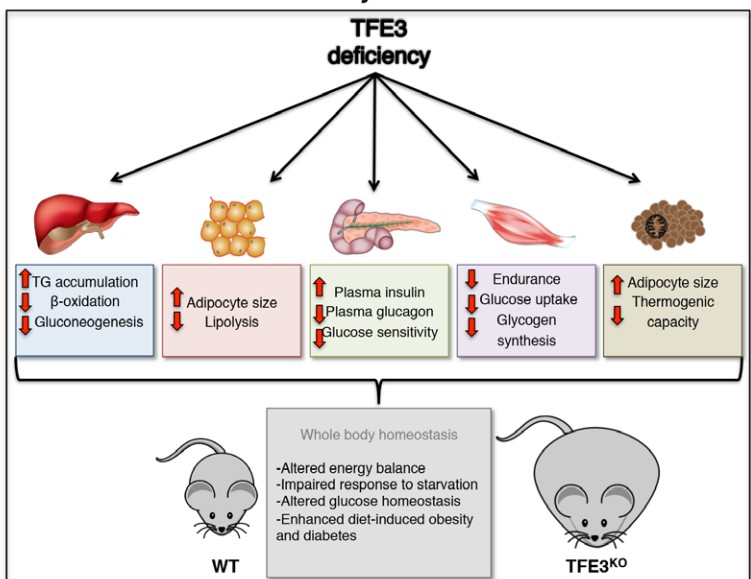

**Figure 9. The role of TFE3 in mitochondrial dynamics and whole-body metabolism.**
Schematic representation of the role of TFE3 in the regulation of mitochondrial dynamics by direct control of *Fis1* expression (left panel) and its contributory role to whole-body metabolism by tissue (right panel).

treated with HFD with the HDAd-PEPCK-*TFE3* virus. Remarkably, viral-mediated TFE3 overexpression in these mice completely compensated for the lack of TFEB and fully rescued the obese phenotype (Fig 7A), liver and body weight (Fig 7F and G) and the accumulation of lipid droplets (Fig 7H). Similarly, TFEB overexpression in the liver of *Tfe3* KO mice treated with HFD also reduced weight gain and the accumulation of lipid droplets (Fig EV4A–C), indicating that the two family members, when overexpressed, are interchangeable in this scenario. In order to understand whether they are cooperative or redundant, we generated double KO (DKO) mice that lack both TFEB and TFE3 in the liver (*Tcfeb* liver KO; *Tfe3* full KO). When fed with a HFD for 12 weeks, DKO mice showed an exacerbated increase in body size (Fig 8A and B), liver weight (Fig 8C) and lipid accumulation compared to the either of the single KO genotypes (Fig 8D). Moreover, PTT analysis showed that the depletion of both TFE3 and TFEB results in an additive reduction in glucose production (Fig 8E).

These results demonstrate that both TFE3 and TFEB are required for the regulation of lipid and glucose homeostasis and that the combined depletion of both transcription factors results in additive alterations in liver metabolism, indicating that they are cooperative rather than redundant.

## Discussion

Previous studies using TFE3 transgenic mice showed that TFE3 is capable of activating genes involved in energy metabolism processes including glycolysis and insulin signalling (Nakagawa *et al*, 2006; Fujimoto *et al*, 2013; Kim *et al*, 2013; Xiong *et al*, 2016). However, as these studies were limited to overexpression, the physiological role of TFE3 has remained elusive. In the present

study, we demonstrate that the absence of TFE3 causes systemic alterations in lipid and glucose metabolism. TFE3 depletion enhances diet-induced obesity and diabetes, while its overexpression has the opposite effects. Interestingly, we found that TFE3, like TFEB (Medina *et al*, 2015), translocates to the nucleus during exercise, suggesting its involvement in exercise-induced adaptive response. Consistently, we observed that TFE3 depletion results in diminished endurance and abolished exercise-induced metabolic benefits. Indeed, 8 weeks of exercise training failed to improve obesity, endurance capacity and glucose homeostasis in *Tfe3* KO mice. This is likely due to the role of TFE3 in glucose homeostasis as *Tfe3* KO mice failed to recover blood glucose and muscle glycogen following exercise.

We have previously reported that liver-specific loss of TFEB exacerbates diet-induced obesity (Settembre *et al*, 2013a). Here we show that TFE3 overexpression rescues the obese phenotype and the glucose intolerance in WT as well as *Tcfeb* LiKO mice fed HFD. TFE3 overexpression increases the expression of genes involved in lipid metabolism and reduces hepatic lipid accumulation. Strikingly, TFEB overexpression is also able to rescue obesity in *Tfe3* KO mice. These results suggest that TFE3 and TFEB have partially redundant roles in cellular metabolism. Given their extensive amino acid sequence similarities and overlapping profiles, it is possible that, in steady-state conditions, TFE3 and TFEB are partially redundant in several tissues and that TFE3 may compensate for TFEB deficiency and *vice versa*. In support of this idea, TFE3 and TFEB have been shown to be mutually redundant transcriptional activators of genes encoding E-cadherin (Huan *et al*, 2005), CD40 ligand (Huan *et al*, 2006), PGC1α (Settembre *et al*, 2013a; Salma *et al*, 2015) and ATF4 (Martina *et al*, 2016), among others. However, in specific physiological contexts, either TFE3 or TFEB may play a more prominent role, a situation similar to that proposed for the differential regulation of

various NFAT subunits (Harris *et al*, 2006). This could explain the strikingly different effects observed between *Tcfeb* KO mice, which die during embryonic development, and *Tfe3* KO mice which are viable. However, under certain conditions that require robust adaptations to environmental cues, such as during starvation, high-fat diet and physical exercise, both TFE3 and TFEB are required and their role becomes cooperative. More studies using conditional KO mice in different tissues and in different conditions are needed to completely decipher the redundant and cooperative roles of these two transcription factors.

Mitochondrial quality control is vital for maintaining cellular and organismal health. Indeed, mitochondrial dysfunction has been implicated in the pathogenesis of several metabolic diseases such as type 2 diabetes, obesity, cardiovascular disease and metabolic syndrome (Petersen *et al*, 2003). Mitochondrial fusion and fission, collectively referred to as mitochondrial dynamics, play an important role in maintaining mitochondrial vitality and thus cellular health (Gomes *et al*, 2011). How nutrients and the cellular metabolic state regulate mitochondrial dynamics is an area of active investigation. Here we show that *Tfe3* null mice, previously described to be indistinguishable from their WT littermates, have severe mitochondrial abnormalities. *Tfe3* deficiency results in accumulation of enlarged and dysfunctional mitochondria with impaired respiration leading to compromised lipid catabolism and exercise intolerance. *Tfe3* null mice also show a reduced expression of genes involved in mitochondrial dynamics and oxidative metabolism, while TFE3 overexpression has the opposite effect.

In summary, our study reveals a central role for TFE3 in the regulation of glucose homeostasis, lipid metabolism and mitochondrial dynamics (Fig 9) and underlines the importance of this transcription factor in the metabolic response to environmental cues and pathological conditions.

# Materials and Methods

## Mouse experiments

*Tfe3* KO mice were previously described (Steingrimsson *et al*, 2002). Conditional *Tcfeb*-flox mouse line generation was previously described (Settembre *et al*, 2012). The liver-specific *Tcfeb* knockout mouse line was generated by crossing the *Tcfeb*-flox mouse line with *Alb*-Cre transgenic mice (Jackson Lab). For all the experiments involving *Tcfeb* LiKO mice, the control mice were *Tcfeb* loxP/loxP mice that did not carry the *Alb-Cre* transgene. All mice used were males (2–3 months old) and maintained on a C57BL/6J strain background. Standard food and water was given *ad libitum*. For fasting experiments, food pellets were removed from the cages for 24 h. To refeed, the food was added to the cage after 24 h of fasting for 3 h. In the high-fat diet study, age-matched male mice were fed *ad libitum* a Western-style diet (Harlan Teklad TD 88137) containing 21% (w/w) total lipid (42% calories as anhydrous milk fat). Body weights were recorded every week. The mice were raised under SPF conditions in the animal facility of the Neurological Research Institute (NRI). All experiments were approved by the Baylor College of Medicine Institutional Animal Care and Use Committee (IACUC) and conform to the legal mandates and federal guidelines for the care and maintenance of laboratory animals. The number of mice used is indicated in each figure. The mice were assigned to the experimental groups based on the genotype.

Mice for muscle electroporation experiments were maintained in a temperature- and humidity-controlled animal care facility, with a 12-h light/dark cycle and free access to water and food. All procedures were formerly approved by the Italian Public Health, Animal Health, Nutrition and Food Safety, Italian Ministry of Health (D.M. No. 75/2014-B).

## *In vivo* metabolic analysis

Analysis of oxygen consumption ($V_{O_2}$), carbon dioxide consumption ($V_{CO_2}$) and respiratory exchange ratio (RER) was performed using the Oxymax Columbus Instruments Comprehensive Lab Animal Monitoring System; mice were acclimatized to the system for 24 h before data collection. $V_{O_2}$ and $V_{CO_2}$ data were normalized to lean mass, while energy expenditure data were normalized to body weight. For glucose tolerance tests, mice were injected with glucose (1.5 mg/g body weight) after 6 h of fasting. For insulin tolerance tests, 4-h-fasted mice were injected (intraperitoneal) with insulin (0.75 milliunit/g body weight for chow diet and 1.0 milliunit/g body weight for HFD, Humulin R; Eli Lily). For pyruvate tolerance test, mice were injected with sodium pyruvate (2.0 g/kg body weight) after 16-h fasting. Experiments were performed between 10:00 and 12:00. Blood was drawn 15, 30, 60 and 120 min after treatment for determination of glucose and insulin levels. Blood glucose concentrations were measured through tail bleed before and at the times indicated after injection. Muscle glucose uptake has been performed as previously described (Zisman *et al*, 2000) with a slight modification. For lipolysis studies, we injected 4-h-fasted mice with IP β3-specific agonist CL316243, or a non-specific β agonist isoproterenol (0.1 mg/kg) and collected blood at 0 min (before injection) and 15 min (after stimulation); we determined plasma glycerol and non-esterified fatty acids (NEFA) levels as a measure of lipolysis *in vivo* (Kyle *et al*, 2016).

## Tissue lipid quantification

Liver triglycerides (TG) were extracted as follows: Briefly, pulverized liver was homogenized in PBS, then extracted using chloroform/methanol (2:1), dried overnight and re-suspended in a solution of 60% butanol 40% Triton X-114/methanol (2:1). Measurements were normalized to protein content in initial homogenate by DC Protein assay (Bio-Rad).

## Blood chemistry analysis

Blood was collected from the retro-orbital plexus under isoflurane (Vedco Saint Joseph, MO) anaesthesia. Serum was frozen at −20°C or used immediately after collection. Specific kits were used for the determination of serum lactate (Nova Biomedical), glucagon (Sigma-Aldrich), leptin, adiponectin and insulin (Millipore). Plasma glucose was monitored by a glucometer.

## Histology

Livers, epididymal fat (eWAT) and brown adipose tissue (BAT) were dissected, fixed with buffered 10% formalin overnight at 4°C

and stored in 70% ethanol, embedded into paraffin blocks and cut into 6- to 10-μm sections. Tibialis anterior (TA) and gastrocnemius muscles were snap-frozen and cut into 10-μm sections. H&E staining was performed following the IHC World protocol. For Oil Red O staining, OCT-embedded tissues were cut into 25-μm sections, fixed in formalin and stained following the IHC World protocol. For periodic acid–Schiff (PAS) staining, the sections were rehydrated in PBS pH 7.4 and stained using the PAS Kit (Sigma-Aldrich, Saint Louis, MO) following the manufacturer's instructions. Immunofluorescence staining was performed on 10-μm cryosections as previously described (Sandri et al, 2004; Mammucari et al, 2007) and then monitored with a fluorescence microscope Leica DM5000B equipped with a Leica DFC300-FX digital CCD camera. For nuclear localization studies, cryosections were stained with mouse-anti-dystrophin-1 IgG2a antibodies (Novocastra) and Hoechst to identify the subsarcolemmal position of myonuclei.

## Liver glycogen content

For measurements of total glycogen in livers, we used the Glycogen Assay Kit (Cell Biolabs, Inc, San Diego, CA) following the manufacturer's protocol.

## Electron microscopy

Liver and muscle specimens were fixed in a mixture of 2% paraformaldehyde and 1% glutaraldehyde prepared in 0.2 M HEPES. Samples were then post-fixed in a mixture of osmium tetroxide and potassium ferrocyanide, dehydrated in ethanol and propylene oxide and embedded in epoxy resin as described previously (Polishchuk et al, 2014). 65-nm-thin sections were cut using a Leica EM UC7 ultramicrotome. EM images were acquired using a FEI Tecnai-12 electron microscope (FEI, Eindhoven, the Netherlands) equipped with a VELETTA CCD digital camera (Soft Imaging Systems GmbH, Munster, Germany). Number of mitochondria was counted using the same magnification within 100-μm square field of view. For each experiment, between 244 and 365 individual mitochondrial diameters were measured from two mice per group.

## Acute and chronic exercise

For acute exercise studies, 8-week-old (WT and KO) mice were randomized into three groups: (i) control sedentary (con), (ii) exercise (Ex) and (iii) exercise with 2 h of recovery (Ex+R). Ex and Ex+R groups were run on a treadmill at 10° uphill, to failure. Mice were subjected to a single bout of running starting at the speed of 5 m/min for 5 min, 10 m/min for 10 and 45 min at 15 m/min, and treadmill speed was then increased at a rate of 1 m/min every 2 min until failure. Failure was defined as the point at which the animals could not continue running after three consecutive shocks. Total running time and distance were recorded. Blood lactate and glucose were collected immediately upon exercise cessation and following a 2-h recovery period (Ex+R only). Animals were euthanized immediately after exercise (Ex) or 120 min post-exercise (Ex+R). Muscle was then excised and frozen in liquid nitrogen for later use. For long-term exercise training, WT and KO mice (8 weeks old) were first acclimatized to the treadmill and tested for

exercise performance (as described in acute exercise) and glucose tolerance. Mice were then randomly divided into four cohorts, including: (i) mice fed a regular chow diet without daily exercise, (ii) mice fed a HFD containing 42% fat without daily exercise, (iii) mice fed a HFD with daily exercise and (iv) mice fed a HFD without daily exercise. Prior to initiation of exercise, mice were fed a HFD for 4 weeks. At the end of the fourth week, all mice were once again tested for performance. Exercise groups were then trained on the treadmill with 10° uphill incline for 50 min/day, 5 days/week at 10 m/min for 8 weeks. Mice were given a HFD during the 8-week training period. Mouse exercise performance and glucose tolerance were tested one last time at the end of the training protocol.

## Animal in vivo transfection experiments

In vivo transfection experiments were performed by intramuscular injection of expression plasmids in tibialis anterior muscle followed by electroporation as previously described (Sandri et al, 2004). Eight days after transfection, mice were exercised on a treadmill. Mice performed exercise on a treadmill (LE 8710 Panlab Technology 2B, Biological Instruments), with a 10° incline, to exhaustion and then were euthanized. Muscle tissue was collected and immediately frozen in liquid nitrogen.

## RNA extraction and quantitative PCR

Total RNA was extracted from tissues in TRIzol reagent (Life Technologies, Carlsbad, CA) using RNeasy kit (Qiagen, Hilden, Germany). RNA was reverse-transcribed using a first-strand complementary deoxyribonucleic acid kit with random primers according to the manufacturer's protocol (Applied Biosystems). The RT–PCRs were performed using CFX96 Real-Time System (Bio-Rad, Hercules, CA). The PCR was performed using the iTaq SYBR Green Supermix (Bio-Rad, Hercules, CA) using the following conditions: pre-heating, 5 min at 95°C; cycling, 40 cycles of 15 s at 95°C, 15 s at 60°C and 25 s at 72°C. Results were expressed in terms of relative fold increase in expression levels as determined by ΔΔCt. Primers used for qPCR are listed in Appendix Table S4. Gapdh, Ribosomal protein S16 and β-actin genes were used as endogenous controls (reference markers).

## Western blotting

Liver samples were homogenized in RIPA buffer (50 mM Tris–HCl pH 7.4, 150 mM NaCl, 1% Triton X-100, 1 mM EDTA pH 8.0, 0.1% SDS) containing Complete protease inhibitor cocktail (Roche Diagnostics, Indianapolis, IN). Samples were incubated for 20 min at 4°C and centrifuged at 16,000 g for 10 min. The pellet was discarded and cell lysates were used for Western blot analysis. Ten to twenty micrograms of liver protein was electrophoresed on a 4-20% SDS–PAGE polyacrylamide gel. After transfer to nitrocellulose or PVDF membrane, the blots were blocked in TBS–Tween-20 containing 5% non-fat milk for 1 h at RT, and then, primary antibody was applied overnight at 4°C. Anti-rabbit IgG or anti-mouse IgG conjugated with horseradish peroxidase (GE Healthcare, Little Chalfont, UK) and ECL (Pierce, Thermo Fisher Scientific, Wilmington, DE) was used for detection. Antibodies used for immunoblots are listed in Appendix Table S5.

## HDAd virus production and injection

HDAd-*TFE3* was generated similarly to HDAd-*TFEB*, as previously described (Pastore *et al*, 2013; Settembre *et al*, 2013a). HDAds were produced in 116 cells with the helper virus AdNG163 as described in detail elsewhere (Ng *et al*, 2002; Palmer & Ng, 2003). Hepatic transduction was achieved by intravenous administration (retro-orbital) of ~200 μl at the dose of $5 \times 10^{12}$ viral particles per kg. Age- and sex-matched mice infected with a transgene deficient HDAd vector served as controls.

## Cellular fractionation

Enriched nuclear and cytosolic cellular subfractions were isolated by differential centrifugation, as previously described (Vainshtein *et al*, 2011). Briefly, the liver was minced on ice and homogenized using a Teflon pestle and mortar and suspended in mitochondrial isolation buffer (MIB; 250 mM sucrose, 20 mM HEPES, 10 mM KCl, 1.5 mM $MgCl_2$, 1 mM EDTA, 1 mM EGTA) supplemented with protease and phosphatase inhibitor cocktails (Complete and PhosSTOP Roche, Roche Diagnostics, Basel, Switzerland). The homogenates were then centrifuged at 1,000 *g* for 10 min at 4°C to pellet the nuclei while mitochondrial and cytosolic fractions were contained within the supernatant. The supernatant fraction was re-centrifuged twice at 16,000 *g* for 20 min at 4°C to pellet the mitochondria and supernate containing cytosolic subfraction was collected. Pellets containing nuclei were re-suspended in nuclear lysis buffer (1.5 mM $MgCl_2$, 0.2 mM EDTA, 20 mM HEPES, 0.5 M NaCl, 20% glycerol, 1% Triton X-100), incubated on ice for 30 min, and then sonicated $3 \times 10$ s followed by a final centrifugation step of 15 min at 16,000 *g*. The supernatant was collected to obtain the enriched nuclear fraction. Fraction purity was determined by Western blot analysis.

## Identification of TFE3 mitochondrial target genes

The TFE3 target genes list was obtained by Betschinger *et al* (2013). The list of well-known mitochondrial genes was downloaded from the Molecular Signatures Database (MSigDB; Liberzon *et al*, 2011; Liberzon, 2014). 136 mitochondrial genes were found to be direct targets of TFE3. The significance of the enrichment of the mitochondrial genes into the TFE3 target genes list was calculated by using the hypergeometric test (*P*-value = 2.38E-23). The analysis of the biological processes was performed by using DAVID online tool (DAVID Bioinformatics Resources 6.7; Dennis *et al*, 2003; Huang da *et al*, 2009).

## Cell culture, plasmids and transfection reagents

Primary mouse hepatocytes were isolated using a two-step perfusion technique as previously described (Seglen, 1976). Briefly, WT or KO mouse liver was perfused with collagenase (C5138, Sigma-Aldrich, Saint Louis, MO) and parenchymal cells were squeezed from the liver. Hepatocyte suspension were further purified by 40% Percoll gradient (P4937, Sigma-Aldrich, Saint Louis, MO), washed with hepatocyte wash medium (17004-024; Thermo Fisher Scientific) and plated in an appropriate cell density. Cells were harvested after 24 h for analysis. For immunofluorescence, cells were fixed in 4% PFA for 20 min, washed twice in PBS, permeabilized in PBS–Triton

0.01% and incubated with the primary antibody overnight at 4°C. The next day, the cells were washed in PBS and incubated with the secondary antibody for 1 h. Coverslips were mounted using the ProLong® Gold Antifade Reagent with DAPI (Invitrogen).

HeLa CRISPR/Cas9 cells were kindly provided by R. Youle (National Institute of Health, NIH) and cultured in Dulbecco's modified Eagle's medium (DMEM) supplemented with 10% foetal bovine serum (FBS), 100 units/ml penicillin and 100 μg/ml streptomycin in 5% $CO_2$ at 37°C. Wild-type and *Tfe3* KO murine embryonic fibroblasts (MEFs) were isolated from embryos at E13.5 following previously established protocols and cultured in DMEM supplemented with 10% FBS, 100 units/ml penicillin and 100 μg/ml streptomycin in 5% $CO_2$ at 37°C.

For ROS assessment, MEFs were transfected with Matrix-roGFP plasmid, a gift from Paul Schumacker (Waypa *et al*, 2010; Addgene plasmid # 49437). Cells were transfected using Lipofectamine LTX reagent (Invitrogen) according to the manufacturer's instructions.

## Mitochondrial membrane potential analyses

Mitochondrial membrane potential was measured in Hela, MEFs and isolated hepatocytes. Mitochondrial membrane potential was measured by epifluorescence microscopy based on the accumulation of TMRM fluorescence as previously described (Romanello *et al*, 2010).

## Seahorse XF-24 metabolic flux analysis

Oxygen consumption was measured at 37°C using an XF-24 extra-cellular analyser (Seahorse Bioscience Inc., North Billerica, MA, USA). Primary hepatocytes ($2 \times 10^4$) were seeded in 24-well plates. After 24 h, the medium was replaced with unbuffered DMEM containing 5 mM glucose and 1 mM pyruvate and the cells were incubated at 37°C without $CO_2$ for 1 h. Substrates were added at a final concentration of oligomycin 1 μM, FCCP 1 μM and antimycin A 4 μM. The experiment was performed in hepatocytes isolated from 4 WT and 4 *Tfe3* KO mice. Each data point represents an average of 10 different wells. Mitochondria were isolated from WT and *Tfe3* KO livers as previously described (Frezza *et al*, 2007). 15 μg of mitochondria were plated on a 24-well plate in MAS buffer (70 mM sucrose, 220 mM mannitol, 5 mM $KH_2PO_4$, 5 mM $MgCl_2$, 2 mM HEPES, 1 mM EGTA, 0.2% fatty acid-free BSA) supplemented with 2 μM rotenone and 10 mM succinate and centrifuged for 20 min at 2,000 *g* at 4°C. Mitochondria were incubated at 37°C without $CO_2$ for 10 min. Substrates were added at a final concentration of ADP 1 mM, oligomycin 2 μM, FCCP 4 μM and antimycin A 4 μM.

## Chromatin immunoprecipitation (ChIP)

ChIP was performed using livers of 2-month-old WT control and WT injected with HDAd-*TFE3* virus 1 month after injection and *Tfe3* KO mice as previously described (Settembre *et al*, 2013a). Primers for a different E-Box not target for TFE3 have been used as negative control. Primers used:

5′-GCCTTTAATCCCAGCAATCA-3′ *Fis1* E-Boxes Forward
5′-ATACCCTCTCCATGCTCCAC-3′ *Fis1* E-Boxes Reverse
5′-TGGTGGTGGTTTGATTTCAT-3′ *Ctrl* E-Box Forward
5′-AAGGTTAGAGGGCTAGACTGAGG-3′ *Ctrl* E-Box Reverse

### The paper explained

#### Problem
The mechanisms by which complex organisms adapt their energy metabolism to environmental cues are not fully understood. Elucidation of such adaptation mechanisms, and identification of the metabolic pathways involved, is of pivotal importance for the understanding of human physiology and disease pathogenesis.

#### Results
We demonstrated that the transcription factors TFE3 and TFEB regulate lipid and glucose metabolism and mitochondrial biogenesis, and play a cooperative role in the control of the adaptive response of whole-body metabolism to environmental cues such as diet and physical exercise.

#### Impact
Our study reveals the importance of TFE3 and TFEB in the metabolic response to environmental cues and in pathological conditions associated with defective glucose and lipid metabolism. Indeed, these transcription factors may represent novel therapeutic targets for diet-induced obesity and diabetes..

### Statistical analyses

Obtained data were processed in Excel (Microsoft Corporation) and Prism (GraphPad Software) to generate bar charts and perform statistical analyses. Data are expressed as mean values ± standard error (SE). Statistical significance was computed using Student's two-tailed *t*-test or ANOVA as reported in the figure legends.

**Expanded View** for this article is available online.

### Acknowledgements

We thank Carmine Settembre and Maxime William C. Rousseaux for critical reading of the manuscript. We are grateful to the technical support of TIGEM Bioinformatic facility. In particular, we thank D. Carrella, A. Carissimo, M. Mutarelli and R. De Cegli. We thank the Vector Core at TIGEM for the help with production of adeno-associated virus (AAV), and Patrizia Annunziata and Nicola Brunetti-Pierri (TIGEM) for HDAd preparation. We thank Kangho Kim (Baylor College of Medicine) for primary hepatocyte isolation and the Mouse Metabolism under ATC and DRC. The project was supported in part by IDDRC grant number 1U54 HD083092 from the Eunice Kennedy Shriver National Institute of Child Health & Human Development. IDDRC Microscopy and Behavioral Cores were used for this project. This work was supported by the US National Institutes of Health (R01-NS078072) and Italian Telethon Foundation (TGM16YCBDM) to A.B., ERC (MyoPHAGY 2823.10) and Leducq to M.S.

### Author contributions

NP and AV performed most of the experiments. TJK performed the ChIP experiment. AA performed the muscle electroporation experiments. TH and NJH provided technical support to NP and AV. EVP performed the EM analysis. MS contributed to the interpretation of the results. AB and NP designed the overall study, supervised the work and wrote the manuscript.

### Conflict of interest

The authors declare that they have no conflict of interest.

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
