## [Review Process File · EMBO Molecular Medicine]

TFE3 regulates whole-body energy metabolism in cooperation with TFEB

Nunzia Pastore, Anna Vainshtein, Tiemo J. Klisch, Andrea Armani, Tuong Huynh, Niculin J. Herz, Elena V. Polishchuk, Marco Sandri and Andrea Ballabio

Corresponding author: Andrea Ballabio, Telethon Institute of Genetics and Medicine

Review timeline:

Submission date:	14 October 2016
Editorial Decision:	22 November 2016
Revision received:	19 December 2016
Editorial Decision:	18 January 2017
Revision received:	30 January 2017
Accepted:	03 February 2017

Transaction Report:

Editor: Roberto Buccione

1st Editorial Decision

22 November 2016

Thank you for the submission of your manuscript to EMBO Molecular Medicine. We have now heard back from the three Reviewers whom we asked to evaluate your manuscript.

Although the Reviewers are positive and agree on the potential interest of the manuscript, they raise a number of important and mostly overlapping general issues.

These mostly centre on the insufficient experimental evidence to support some of the main conclusions, including insulin sensitivity, energy expenditure/balance, etc. In essence, as reviewer 3 also mentions, we do not really learn the origin of the obesity phenotype. Reviewer 1 also notes the tendency to overstate and a number of technical issues that need to be resolved. I will not dwell into much detail, but I would like to highlight the main specific points.

Reviewer 1 notes that the phenotyping of glucose metabolism is insufficiently developed as is energy balance assessment, which compromises the ability to conclude that TFE3 plays a key role in whole body glucose homeostasis. This reviewer also lists a number of important technical concerns.

Reviewer 2 is less reserved but similarly to reviewer 1, expresses concerns on the conclusions deriving from the energy expenditure data. S/he would also like you to better analyse the liver phenotype, and would like you to improve the analysis of FAO gene expression in the KO and overexpressor mice. Reviewer 2 also mentions a few other action points.

Reviewer 3 essentially recapitulates the concerns mentioned by both reviewer 1 and 2.

After further reviewer cross-commenting and internal discussion, it was agreed that the further experimentation is both warranted and feasible. It was noted that perhaps the insulin clamp approach (ideal) might require a not readily available expertise, but at least GTT and ITT on chow with insulin assessment during the GTT should be performed. Furthermore, it was appreciated that an additional cohort of mice would probably be required to perform the experiments related to energy balance assessment and liver metabolism. In fact, it was reiterated that to address why the mice are obese is important.

In conclusion, while publication of the paper cannot be considered at this stage, given the potential interest of your findings and after internal discussion, we have decided to give you the opportunity to address the criticisms.

We are thus prepared to consider a substantially revised submission, with the understanding that the Reviewers' concerns must be addressed with additional experimental data where appropriate and as outlined above, and that acceptance of the manuscript will entail a second round of review. The overall aim is to significantly upgrade the relevance and conclusiveness of the dataset, which of course is of paramount importance for our title.

I look forward to seeing a revised form of your manuscript in due time.

***** Reviewer's comments *****

Referee #1 (Remarks):

The manuscript by Pastore and colleagues contains interesting information concerning the physiological role of TFE3 and its relationship with TFEB. They found that deletion of TFE3 results in altered mitochondrial morphology and function in mice and cells. They also found that TFE3 KO mice have altered energy and glucose homeostasis, especially when challenged a high-fat diet. They also show that viral-mediated TFE3 liver/muscle overexpression improves the metabolic abnormalities of the total KO. Authors concluded that TFE3 and TFEB play a cooperative role in response to metabolic challenges by providing convincing data that dual depletion results in additive effects. The text is clear and well written. Most of the experiments support the conclusion claimed by the authors, especially in regards to mitochondria dynamics. However, I have some concerns that need to be addressed in regards to the regulation of glucose and energy metabolism.

Major points

- 1) While the studies of mitochondrial dynamics and function are well performed and supported by comprehensive data, phenotyping of glucose metabolism is weak. There is no assessment of insulin sensitivity and glucose tolerance in chow-fed mice and the absence of insulin clamp, the gold standard for assessing insulin action, preclude any robust conclusions in regard to liver or muscle insulin sensitivity. Insulin and glucagon levels should also be presented to support the differences in glycemia. The methods section indicates that blood was sampled for insulin during the GTT, but these data are not presented.
- 2) Energy balance assessment also needs a better characterization. There is no indication about the settings in which food intake was recorded. Mice fed a chow diet normally eat about 4g per day and Fig S1 reported values of 0.07g. Were animals grouped? Was it taken randomly during the day? Were animals fasted before? How old were they? Were they eating the same amount of food during the light and dark cycles? The average RER should be measure for both light and dark cycles (Fig 1E). In addition, oxygen consumption data, as an indirect representation of energy expenditure, should be presented. The observation that TFE3 KO mice fed a high-fat diet gained significantly more weight need to be justified by changes in energy balance. Since authors argue that there is no change in food intake, there should be at least changes in one of these parameters: thermogenesis, activity, or fecal energy loss. Evaluation of brown adipose tissue thermogenic capacity and/or activity (by looking at oxidative genes and uncoupling protein expression) should also be included to support the data.
- 3) When fed a chow diet, TFE3 KO mice exhibit a 50% increase in fat mass. Since there is a strong

relation between fat distribution and glucose metabolism, measurement of fat distribution or at least comparison of the weight of visceral versus subcutaneous fat depots should be included. In addition, it is unclear which visceral fat was considered for Fig 2E? These data should also be expressed in absolute value rather than in % to body weight. In addition, absolute quantification of liver triglycerides should be included to the manuscript.

4) Authors have this tendency to overstate their conclusions. By example, when saying that higher blood lactate indicates that mice utilize anaerobic glycolysis as a source of energy. It definitely may suggest it but it could also have other significations. Another example is when stating that these data clearly indicate that TFE3 plays a key role in regulating whole body glucose homeostasis, which conclusion requires additional experiments, as indicated in my previous comments. Again, conclusion that food intake is not different is not well supported as there is no systematic assessment of food intake over time. Other examples include these statements that referred to non significant data: 1) mice injected with HDAd-PEPCK-hTFE3 showed increased in genes involved in lipid catabolism, 2) Tfe3 KO mice demonstrated increased plasma FFA, 3) Glut4 expression was reduced. In addition, PEPCK is not liver specific like what is stated. It is also highly express in other tissues such as white and brown fat, kidneys and adrenals.

5) I also have some concerns with the ChIP experiments. Additional controls for transcriptional factors known to bind/not bind the same region of the promoter should be included. In addition, at least one primer set outside of the binding regions should be included to show an absence of enrichment. In addition, the primer sets for E-box1 and E-box2 overlap (-374,-528 and -475,-635) which may explain why the enrichment is similar.

6) The Seahorse experiment data should be corrected for protein or DNA content in each well, as the differences in basal respiration may suggest difference in the number of cells. In addition, the number of plated cells, the composition of the media (glutamate, pyruvate), the time without CO₂ and the number of time the experiment was repeated should be included. Isolated mitochondria from liver could also add interesting information to the manuscript, as primary hepatocytes are highly metabolic.

7) The number of subjects for each figure should be clearly indicated.

8) Authors used floxed mice as controls. Did they validate whether the Alb-Cre transgene per se could influence glucose metabolism? Normally, Cre positive animals should also be included as controls.

9) Since several reports indicate huge differences between C57BL/6J and 6N mice in regards to glucose homeostasis, authors need to clarify the specific background of the animals used.

10) Glyceraldehyde 3-phosphate dehydrogenase as a housekeeping genes sound a bad choice for the actual experimental settings. Please indicate the ct values of Gapdh for each groups.

Minor comments

11) In Fig 2 and S1, the weight of animals and tissues should be reported in absolute value. Average RER should be presented for both light and dark phases.

12) The PAS staining presented in Fig S1 is hard to interpret. Better resolution images or absolute quantification of glycogen content should be added.

13) In Fig 3, what % of muscular cells expressed GFP? The figure seems to indicate a very low number of cells for the important phenotype reported.

14) In Fig 3, which WAT depots are reported? The method section should explicitly mention whether epididymal, retro-peritoneal, inguinal or others were weighted. 3g for a single depot appears high.

15) The fed TFE3 KO group is missing in Fig S2C.

- 16) Genes presented in Fig S2E are not necessarily involved in oxidation/catabolism of lipids.
- 17) In Fig S3, the small difference in leptin levels does not seem to correlate with the huge difference in fat mass.
- 18) There is no reference to Fig S5 in the text.
- 19) References to Fig 4E and 4F at the end of the last paragraph of the mitochondrial dynamics results section don't fit with the figure.
- 20) The last paragraph of the results section referred to Fig EV5A-C, which I can't find.
- 21) Were mice injected with pyruvate or sodium pyruvate for the PTT?
- 22) ROS levels in tissues could be measure to support the reduction in complex I activity.

Referee #2 (Remarks):

In their Manuscript, Pastore and colleagues explore the physiological roles of TFE3, a HLH-leucine zipper transcription factor member of the MiT family. Through a series of well-designed and appropriately executed studies, the authors describe an unexpected complex phenotype in Tfe3 KO mouse, characterized by impaired mitochondrial function, altered energy balance and altered glucose and lipid metabolism, which predisposed animals to obesity and diabetes when fed a high-fat diet. Moreover, the authors also manage to demonstrate that TFE3 and TFEB, another member of the MiT family that has considerable target overlap with TFE3, have additive rather than redundant roles in the control of metabolism responses to diet or exercise.

However, this Reviewer believes a few points should be addressed prior to publication of the paper.

1. Tfe3 KO mice present hepatic alterations, in particular an apparent increase in lipid deposition, probably due to defects in mitochondrial function. However, given the importance of this point, a better characterization of hepatic steatosis should be performed. For example, a direct quantification of total liver lipid content would be much more informative than a measurement derived from the oil red staining of liver sections, which may not accurately reflect the situation of the whole liver.
2. The authors state that the expression of genes involved in lipid metabolism is reduced in livers from Tfe3 KO mice, however these genes are not detailed in the text. Moreover, a more profound analysis of expression of genes involved in fatty acid oxidation (such as Cpt1) or in lipogenesis (such as Fas or Srebp1c) should be performed in KO and overexpressing mice.
3. Since Tfe3 KO mice present changes in adiposity and mitochondrial function, one would expect these changes to be reflected in changes in energy expenditure. In this aspect, only RER is reported. The quantification of energy expenditure should also be displayed.
4. In order to fully appreciate the quality and solidness of the data, the number of animals used for each test has to be indicated.
5. In the Results section, the authors state that "in vivo-stimulated lipolysis was impaired...". It is necessary to indicate which compound was used to stimulate lipolysis. In addition, FFA may not be the best product of lipolysis to measure since FFA may be re-esterified. For this reason, glycerol is considered the best indicator of lipolysis. In the Methods section it is stated that glycerol levels have been measured, however, values are not displayed. This should be clarified.

Minor comments:

- In the main text, page 8, there is mislabeling of Figure S4B and C.
- Page 9, lane 11, it should be Fig 5E instead of Fig 4E.
- Page 9, line 13, it should be Fig 5F instead of Fig 4F.
- Page 10, line 10, it should be Fig S5A-C instead of Fig EV5A-C

Referee #3 (Remarks):

The paper studies the phenotype of Tfe3 in systemic metabolism and hepatic function. The authors show that global Tfe3 ablation results in increased adiposity which exacerbated on a HFD. Loss of Tfe3 in liver leads to altered hepatic mitochondrial function and thereby the disability to utilize lipids. Furthermore, the authors demonstrate that Tfe3 and Tfeb seem to act in a concerted manner. The paper is very well done and the data convincing. The clarity of the data needs to be improved somewhat since it's a complex metabolic phenotype which leads to altered exercise capacity and obesity and the reason for this remains a bit unclear.

One major point is that we do not know where the obesity phenotype is coming from. Is this reduced lipid metabolism in the liver? If yes a few points of data should be added to support this notion. For example is O₂ and CO₂ changed. The authors only present RER which demonstrates altered substrate utilization but this does not explain where the energy imbalance is coming from.

Also, the data on exercise needs to be better integrated as it is difficult to discern between the two phenotypes. Why would altered lipid utilization in the liver prevent the exercise effect. Is this because of a mixture of phenotypes coming from muscle and liver in the global ko? In that regards the data from reduced Glut4 mRNA is not convincing. If the authors want to investigate the phenotype of muscle function in more details (which I don't think is necessary given the liver data) they would need to perform more in depth studies such as Glut4 protein and localization as well as muscle insulin sensitivity measurements.

Other points:

Why is 2D showing different results compared to 3E. In 2D there is a difference in weight gain in 3E there is not. I cannot see where these differences would come from.

The data from Fig. 3 is overinterpreted. Exercise also improves Tfe3 mice function. The liver weight data is quite similar and the fact that it not significant is mainly due to the much larger error bars. The effect on AT is clearly different.

Minor points:

Fig. 2D please show actual weight not percent weight gain

Fig. 3E please also show AT for Fig 3D to support the data from Fig. 3G

1st Revision - authors' response

19 December 2016

To the Editor:

We thank the editor for considering our manuscript for publication as research article in *Embo Molecular Medicine* and the reviewers for their positive feedback and comments, as well as for the useful recommendations to improve the manuscript. We addressed the reviewers' concerns, by performing new experiments and/or by modifying the text.

Below is a point-by-point response to the reviewers' comments.

Referee #1 (Remarks):

Major points

1) While the studies of mitochondrial dynamics and function are well performed and supported by comprehensive data, phenotyping of glucose metabolism is weak. There is no assessment of insulin sensitivity and glucose tolerance in chow-fed mice and the absence of insulin clamp, the

gold standard for assessing insulin action, preclude any robust conclusions in regard to liver or muscle insulin sensitivity. Insulin and glucagon levels should also be presented to support the differences in glycemia. The methods section indicates that blood was sampled for insulin during the GTT, but these data are not presented.

We agree with the reviewer's comment. We performed additional experiments for the assessment of glucose metabolism in chow-fed mice. We analyzed insulin and glucose tolerance by ITT (**new Fig 2D**) and GTT (**new Fig 2E**) assays, respectively, and measured insulin secretion during GTT (**new Fig 2F**). These results showed altered glucose sensitivity in chow diet fed mice, while insulin sensitivity was normal. As suggested by the reviewer, we also included insulin (**new Fig 2B**) and glucagon (**new Fig 2C**) levels, which further support an alteration of glucose metabolism.

2) Energy balance assessment also needs a better characterization. There is no indication about the settings in which food intake was recorded. Mice fed a chow diet normally eat about 4g per day and Fig S1 reported values of 0.07g. Were animals grouped? Was it taken randomly during the day? Were animals fasted before? How old were they? Were they eating the same amount of food during the light and dark cycles? The average RER should be measure for both light and dark cycles (Fig 1E). In addition, oxygen consumption data, as an indirect representation of energy expenditure, should be presented. The observation that TFE3 KO mice fed a high-fat diet gained significantly more weight need to be justified by changes in energy balance. Since authors argue that there is no change in food intake, there should be at least changes in one of these parameters: thermogenesis, activity, or fecal energy loss. Evaluation of brown adipose tissue thermogenic capacity and/or activity (by looking at oxidative genes and uncoupling protein expression) should also be included to support the data.

We agree with the reviewer that more data on energy balance would strengthen our findings. We corrected the values for food intake (**new Fig 1C and EV3A**), which was measured single-cage housed 2-month-old mice kept for three days in chow diet. We also included the RER plot and average for both light and dark cycles (**new Fig 1F-G and 3C-D**) as well as oxygen consumption and energy expenditure data, as measured by CLAMS analysis (**new Fig EV1A-B, 1H-I and EV3C-D, 3E-F**). *Tfe3* KO mice fed with a high-fat diet showed reduced energy expenditure (**new Fig 3F and 3G**), which accounts, at least in part, for the obese phenotype in the absence of changes in food intake. We also detected a reduction in *Ucp1*, *Ucp3* and *Ppara* mRNA levels (**new Appendix Fig S1A**) and of UCP1 protein levels (**new Appendix Fig S1B**) in brown adipose tissue from HFD fed TFE3 KO mice compared to WT controls.

3) When fed a chow diet, TFE3 KO mice exhibit a 50% increase in fat mass. Since there is a strong relation between fat distribution and glucose metabolism, measurement of fat distribution or at least comparison of the weight of visceral versus subcutaneous fat depots should be included. In addition, it is unclear which visceral fat was considered for Fig 2E? These data should also be expressed in absolute value rather than in % to body weight. In addition, absolute quantification of liver triglycerides should be included to the manuscript.

We analyzed epididymal (eWAT) and inguinal (iWAT) fat in *Tfe3* KO and WT mice in chow diet. As shown in **new Fig 1D**, we found a significant increase in eWAT and iWAT in *Tfe3* KO mice compared to controls. As suggested by the reviewer, we corrected the figures by indicating which visceral fat we considered and expressed the tissue weight in absolute value (**new Fig 1D**). We also included absolute quantification of liver triglycerides for both chow and high-fat diet fed mice (**new Fig EV2D and 3H**).

4) Authors have this tendency to overstate their conclusions. By example, when saying that higher blood lactate indicates that mice utilize anaerobic glycolysis as a source of energy. It definitely may suggest it but it could also have other significations. Another example is when stating that these data clearly indicate that TFE3 plays a key role in regulating whole body glucose homeostasis, which conclusion requires additional experiments, as indicated in my previous comments. Again, conclusion that food intake is not different is not well supported as there is no systematic assessment of food intake over time. Other examples include these statements that referred to non significant data: 1) mice injected with HDAd-PEPCK-hTFE3 showed increased in genes involved in lipid catabolism, 2) Tfe3 KO mice demonstrated increased plasma FFA, 3)

Glut4 expression was reduced. In addition, PEPCK is not liver specific like what is stated. It is also highly express in other tissues such as white and brown fat, kidneys and adrenals.

We thank the reviewer for these comments. We modified the text accordingly.

5) I also have some concerns with the ChIP experiments. Additional controls for transcriptional factors known to bind/not bind the same region of the promoter should be included. In addition, at least one primer set outside of the binding regions should be included to show an absence of enrichment. In addition, the primer sets for E-box1 and E-box2 overlap (-374,-528 and -475,-635) which may explain why the enrichment is similar.

Our ChIP experiment demonstrated that TFE3 binds the E-boxes in the *Fis1* promoter in WT and in TFE3 overexpressing livers. This binding is lost in KO samples, suggesting specificity of the binding. We included a primer set outside of the binding region as negative control for the enrichment (**new Fig 5G**). It is very hard to design non-overlapping primers as the two E-boxes are very close to each other. Therefore, we amplified the 200bp fragment containing both E-boxes (**new Fig 5G**).

6) The Seahorse experiment data should be corrected for protein or DNA content in each well, as the differences in basal respiration may suggest difference in the number of cells. In addition, the number of plated cells, the composition of the media (glutamate, pyruvate), the time without CO2 and the number of time the experiment was repeated should be included. Isolated mitochondria from liver could also add interesting information to the manuscript, as primary hepatocytes are highly metabolic.

We performed additional experiments and normalized the data for protein content in each well (**new Fig 6E and 6F**) and modified the text accordingly. We also performed OCR measurement in isolated mitochondria (**new Fig 6G and 6H**).

7) The number of subjects for each figure should be clearly indicated.

The number of subjects was included in figure legends.

8) Authors used floxed mice as controls. Did they validate whether the Alb-Cre transgene per se could influence glucose metabolism? Normally, Cre positive animals should also be included as controls.

Several studies indicate that the Alb-Cre transgene has no effects on glucose metabolism (as an example see reference ((Postic et al, 1999)).

9) Since several reports indicate huge differences between C57BL/6J and 6N mice in regards to glucose homeostasis, authors need to clarify the specific background of the animals used.

We used C57BL/6J. This information was included in the background in methods.

10) Glyceraldehyde 3-phosphate dehydrogenase as a housekeeping genes sound a bad choice for the actual experimental settings. Please indicate the ct values of Gapdh for each groups.

We always normalize mRNA levels for three housekeeping genes (*Gapdh*, β -Actin and *SI6*). However, we wrongly reported only *Gapdh* in the methods. We have now included all three housekeeping genes in the methods section.

Minor comments

11) In Fig 2 and S1, the weight of animals and tissues should be reported in absolute value. Average RER should be presented for both light and dark phases.

We modified the figures as suggested (**new Fig 3B and 1D**).

12) The PAS staining presented in Fig S1 is hard to interpret. Better resolution images or absolute quantification of glycogen content should be added.

We have now included better images and the absolute quantification of glycogen in the liver (**new Fig EV1E and EV1F**).

13) In Fig 3, what % of muscular cells expressed GFP? The figure seems to indicate a very low number of cells for the important phenotype reported.

The percentage of GFP expressing cells in electroporated muscles is low but sufficient to evaluate TFE3 nuclear translocation upon acute exercise.

14) In Fig 3, which WAT depots are reported? The method section should explicitly mention whether epididymal, retro-peritoneal, inguinal or others were weighted. 3g for a single depot appears high.

We reported the epididymal fat (eWAT). Methods and figure legends have been modified accordingly. Mice in the experiment of figure 3G (**new Fig 4G**) were fed a HFD for 14 weeks.

15) The fed TFE3 KO group is missing in Fig S2C.

We have now added the fed *Tfe3* KO to the figure as suggested (**new Fig EV2E**).

16) Genes presented in Fig S2E are not necessarily involved in oxidation/catabolism of lipids.

We have now included *Cpt1a* as an example of genes involved in fatty acid oxidation. The text was modified accordingly (**new Fig EV2E**).

17) In Fig S3, the small difference in leptin levels does not seem to correlate with the huge difference in fat mass.

We agree with the reviewer: the difference in leptin levels is not huge, however, it is still significant (1.5-fold).

18) There is no reference to Fig S5 in the text.

19) References to Fig 4E and 4F at the end of the last paragraph of the mitochondrial dynamics results section don't fit with the figure.

20) The last paragraph of the results section referred to Fig EV5A-C, which I can't find.

We fixed the text according to the reviewer's comments.

21) Were mice injected with pyruvate or sodium pyruvate for the PTT?

Mice were injected with sodium pyruvate. We included this information in the methods.

22) ROS levels in tissues could be measure to support the reduction in complex I activity.

We did not measure ROS levels directly, but reported the oxyblot, a measure of carbonyl groups introduced into proteins by oxidative reactions, in livers (**new Appendix Fig S2A**).

Referee #2 (Remarks):

1. *Tfe3* KO mice present hepatic alterations, in particular an apparent increase in lipid deposition, probably due to defects in mitochondrial function. However, given the importance of this point, a better characterization of hepatic steatosis should be performed. For example, a direct quantification of total liver lipid content would be much more informative than a measurement derived from the oil red staining of liver sections, which may not accurately reflect the situation of the whole liver.

We have now included data on total liver lipid content in livers of mice fed a chow diet, fasted and fed a high-fat diet to support the Oil-Red staining data (**new Fig EV2D and 3H**).

2. The authors state that the expression of genes involved in lipid metabolism is reduced in livers from *Tfe3* KO mice, however these genes are not detailed in the text. Moreover, a more profound analysis of expression of genes involved in fatty acid oxidation (such as *Cpt1*) or in lipogenesis (such as *Fasn* or *Srebp1c*) should be performed in KO and overexpressing mice.

We have now included data on *Cpt1a* expression as an example of a gene involved in fatty acid oxidation, and of the *Fasn* and *Srebp1c* genes, which are involved in lipogenesis in *Tfe3* KO and overexpressing livers (**new Fig EV2E**).

3. Since *Tfe3* KO mice present changes in adiposity and mitochondrial function, one would expect these changes to be reflected in changes in energy expenditure. In this aspect, only RER is reported. The quantification of energy expenditure should also be displayed.

We have now included the energy expenditure data for both chow and high-fat diet fed mice (**new Fig 1H-I and 3F-G**).

4. In order to fully appreciate the quality and solidness of the data, the number of animals used for each test has to be indicated.

We have included the number of mice in the figure legends.

5. In the Results section, the authors state that "in vivo-stimulated lipolysis was impaired...". It is necessary to indicate which compound was used to stimulate lipolysis. In addition, FFA may not be the best product of lipolysis to measure since FFA may be re-esterified. For this reason, glycerol is considered the best indicator of lipolysis. In the Methods section it is stated that glycerol levels have been measured, however, values are not displayed. This should be clarified.

We have now included the information on the compound used to stimulate lipolysis in the method section. We have also measured glycerol as a product of lipolysis, as suggested (**new Fig EV3K**).

Minor comments:

- In the main text, page 8, there is mislabeling of Figure S4B and C.

- Page 9, lane 11, it should be Fig 5E instead of Fig 4E.

- Page 9, line 13, it should be Fig 5F instead of Fig 4F.

- Page 10, line 10, it should be Fig S5A-C instead of Fig EV5A-C

We have fixed the text according to the reviewer's indications.

Referee #3 (Remarks):

One major point is that we do not know where the obesity phenotype is coming from. Is this reduced lipid metabolism in the liver? If yes a few points of data should be added to support this notion. For example is O₂ and CO₂ changed. The authors only present RER which demonstrates altered substrate utilization but this does not explain where the energy imbalance is coming from.

We thank the reviewer for these useful comments. We have now included the VO₂ and CO₂ data (**new Fig EV1A-D and EV3C-F**) as well as the energy expenditure data for both chow and high-fat diet fed mice (**new Fig 1H-I and 3F-G**).

Also, the data on exercise needs to be better integrated as it is difficult to discern between the two phenotypes. Why would altered lipid utilization in the liver prevent the exercise effect. Is this because of a mixture of phenotypes coming from muscle and liver in the global ko? In that regards the data from reduced *Glut4* mRNA is not convincing. If the authors want to investigate the phenotype of muscle function in more details (which I don't think is necessary given the liver data) they would need to perform more in depth studies such as *Glut4* protein and localization as well as muscle insulin sensitivity measurements.

The effect that we observed in *Tfe3* KO mice during exercise is likely due to a combination of the liver and muscle phenotypes. Reduced endurance and impaired recovery after exercise are associated also to a reduced gluconeogenesis and increased glucose consumption due to lipid metabolism impairment in the liver. We agree with the reviewer that the *Glut4* data are not convincing and would require more studies. Therefore, we removed these data from the paper.

Other points:

Why is 2D showing different results compared to 3E. In 2D there is a difference in weight gain in 3E there is not. I cannot see where these differences would come from. The data from Fig. 3 is overinterpreted. Exercise also improves Tfe3 mice function. The liver weight data is quite similar and the fact that it not significant is mainly due to the much larger error bars. The effect on WAT is clearly different.

We agree with the reviewer that the effects of exercise on body weights reported in the Fig3E (new Fig 4E) are not convincing. This is likely due to the fact that the body weights in the 4 groups at the beginning of the experiment were variable. We have now reported the data as % on body weight at T0 for each mouse. The difference in weight gain between WT and KO fed with HFD is now clear. *Tfe3* KO mice also clearly show a reduced recovery after exercise (new Fig 4E).

Minor points:

Fig. 2D please show actual weight not percent weight gain

We modified the figure accordingly (new Fig 3B). For the studies with virus injections as well as for the DKO studies we have used the percent weight gain due to the variability in the weight at T0 of the different experimental groups.

Fig. 3E please also show WAT for Fig 3D to support the data from Fig. 3G

We showed the WAT from WT and *Tfe3* KO mice fed with HFD in new Fig 3A. In the new Fig4D we focused on the differences between HFD+exercise WT and *Tfe3* KO mice. Weight data for both groups are reported in the new Fig4G.

References

Postic C, Shiota M, Niswender KD, Jetton TL, Chen Y, Moates JM, Shelton KD, Lindner J, Cherrington AD, Magnuson MA (1999) Dual roles for glucokinase in glucose homeostasis as determined by liver and pancreatic beta cell-specific gene knock-outs using Cre recombinase. *The Journal of biological chemistry* **274**: 305-315

2nd Editorial Decision

18 January 2017

Thank you for the submission of your revised manuscript to EMBO Molecular Medicine and apologies for the delay in providing you with a decision due to delayed delivery of the evaluations in connection with the holiday season.

We have now received the enclosed reports from the reviewers that were asked to re-assess it. As you will see, while reviewer 2 is now completely satisfied, the other two would like you, at the very least, to discuss a number of unclear conclusions especially in relationship to the respiration exchange ratio (RER) data, and missing elements, and also ask for additional parameters (e.g. animal weights) to consolidate the data and increase transparency. Reviewer 1 also lists a few other minor items for you to deal with (more on this further below).

Please carefully discuss each point and provide the missing data. Depending on the completeness of your response, I am willing make an editorial decision on your manuscript. Please make sure the changes are marked up in the new version.

In the event of a positive outcome, there are a number of editorial requirements for you to comply

with before we can proceed with acceptance. I suggest you do so for your next, final revision to reduce manuscript back and forth with the editorial office. The requested amendments are as follows:

- 1) We noted, as did Reviewer 1, that on page 5 of the manuscript you call out Fig EVI and EVJ, Please correct. On the other hand, the reviewer also notes an issue with Appendix Fig. S1B, which we could not reproduce. Please double check to make sure it is OK.
- 2) Please provide a running title
- 3) As per our Author Guidelines, the description of all reported data that includes statistical testing must state the name of the statistical test used to generate error bars and P values, the number (n) of independent experiments underlying each data point (not replicate measures of one sample), and the actual P value for each test (not merely 'significant' or 'P < 0.05').
- 4) The manuscript must include a statement in the Materials and Methods identifying the institutional and/or licensing committee approving the experiments, including any relevant details (like how many animals were used, of which gender, at what age, which strains, if genetically modified, on which background, housing details, etc). We encourage authors to follow the ARRIVE guidelines for reporting studies involving animals. Please see the EQUATOR website for details: <http://www.equator-network.org/reporting-guidelines/improving-bioscience-research-reporting-the-arrive-guidelines-for-reporting-animal-research/>. Please make sure that ALL the above details are reported. Furthermore since you mention Baylor College regulations, we are assuming that all animal experimentation was performed in the US. If this is not correct, please amend appropriately.

Please submit your revised manuscript within two weeks. I look forward to seeing a revised form of your manuscript as soon as possible.

***** Reviewer's comments *****

Referee #1 (Remarks):

Authors performed additional experiments and addressed most of my concerns. They have substantially improved the manuscript. I believe that the section on mitochondrial dynamics the most well-performed and conclusive part of the story. However, there are still issues with the interpretation of metabolic data that need to be addressed, or at least discussed, before the manuscript reaches the standards of EMBO:

Calorimetry data should be expressed in absolute values, as there are important differences in body composition. The differences in lean mass in chow-fed animals (about 3g) and in fat mass in HF-fed animals (about 8g) represent important confounding factors for such analysis. There should also be more emphasis on the important difference in RER between HF-fed WT and KO animals, as this represents a very important observation that may explain the obesity-prone phenotype.

The difference in RER between light/dark phases is highly attributable to food intake and very little to a shift from carbs to FA consumption as stated.

It is very surprising that RER is similar in chow and HF-fed animals (about 0.9). HFD should normally reduce RER to ~0.7.

Insulin levels during GTT should be expressed in absolute values as well, especially considering that basal insulin is higher in TFE3KO mice.

In FigEV3I, the fall in glycemia following insulin administration is massive and almost impossible for WT mice fed a HF diet for 8 weeks (from 210mg/dl to 40mg/dl; severe hypoglycemia). Actually, if we compare to the ITT in Fig2D (chow-fed), HF-fed animals are more insulin sensitive than chow-fed. Was the dose of 0.75U/kg also used for HF-fed animals as reported in the methods section?

Minor comments:

Authors are still overstating by referring to non-significant results:

- P4 "adiponectine levels were reduced"
- P6 "a small increase in plasma FFA"

There are still figures (EVI and EVJ, P5) that are not included.

There is only n=1 reported for Fig4C.

In appendix FigS1A, is the WT line at Y=1 referring to WT, chow-fed animals?

There is a black rectangle over the blot presented in appendix FigS1B that prevents the appreciation of the results.

Referee #2 (Remarks):

The authors have appropriately addressed all my comments

Referee #3 (Remarks):

The revision have addressed all of my concerns except for one point. The authors MUST present the weight data for Fig 4E, 7B and G as well as 8A as absolute values as well. I cannot understand how a group variability could impact the conclusion if there is a 60% or higher body weight gain. These Figures can be presented in the supplemental material but they should be accessible for the reader to asses the data.

2nd Revision - authors' response

30 January 2017

Referee #1 (Remarks):

1) Calorimetry data should be expressed in absolute values, as there are important differences in body composition. The differences in lean mass in chow-fed animals (about 3g) and in fat mass in HF-fed animals (about 8g) represent important confounding factors for such analysis.

We agree with the reviewer that differences in lean and fat mass play a role in the calorimetry data. In the new version of the manuscript we have now represented in the **Appendix Figures 1 and 2** the absolute values of energy expenditure for both chow and HFD fed mice. These data show a significant difference in EE in chow diet (**Appendix Fig 1A and 1B**), but not in HFD (**Appendix Fig 2A and 2B**). This can be explained by the striking difference in body weight between WT and *Tfe3* KO mice at the time of the analysis (WT 34.98±1.5; *Tfe3* KO 43.78±1.9 Student's *t*-test *P*=0.0064). Normalization of calorimetry data allows for better comparison of metabolic rates between subjects of varying size and body composition.

2) There should also be more emphasis on the important difference in RER between HF-fed WT and KO animals, as this represents a very important observation that may explain the obesity-prone phenotype. The difference in RER between light/dark phases is highly attributable to food intake and very little to a shift from carbs to FA consumption as stated.

We modified the text accordingly.

3) It is very surprising that RER is similar in chow and HD-fed animals (about 0.9). HFD should normally reduce RER to ~0.7.

The mice used for calorimetry studies were fed a HFD diet for 4 weeks. A reduction of the RER to 0.7 maybe observed after a longer period of HFD feeding.

4) Insulin levels during GTT should be expressed in absolute values as well, especially considering that basal insulin is higher in TFE3KO mice.

We modified the figures accordingly.

5) In FigEV3I, the fall in glycemia following insulin administration is massive and almost impossible for WT mice fed a HF diet for 8 weeks (from 210mg/dl to 40mg/dl; severe hypoglycemia). Actually, if we compare to the ITT in Fig2D (chow-fed), HF-fed animals are more insulin sensitive than chow-fed. Was the dose of 0.75U/kg also used for HF-fed animals as reported in the methods section?

The reviewer is right. The dose used for the two groups was different: for the chow diet mice we used a dose of 0.75 U/Kg, while for the HFD we used a dose of 1.0 U/Kg as the dose of 0.75U/Kg in HFD fed mice was not sufficient since they were already insulin resistant. The methods section was modified accordingly. We would also like to clarify that the mice used for the ITT were fed a HFD for 4 weeks (not 8 weeks).

Minor comments:

1) Authors are still overstating by referring to non-significant results:

- P4 "adiponectin levels were reduced"

- P6 "a small increase in plasma FFA"

We agree with the reviewer and revised the text accordingly.

2) There are still figures (EVI and EVJ, P5) that are not included.

We thank the reviewer for noticing this error. We modified the figure labels in the text.

3) There is only n=1 reported for Fig4C.

We analyzed n=4. We reported the quantification in **Fig4D** and the relative immunoblots in the source data.

4) In appendix FigS1A, is the WT line at Y=1 referring to WT, chow-fed animals?

The WT line is related to WT mice fed a chow-diet. We included this information in the figure legend.

5) There is a black rectangle over the blot presented in appendix FigS1B that prevents the appreciation of the results.

We improved the quality of **FigS1B**.

Referee #3 (Remarks):

The revision has addressed all of my concerns except for one point. The authors MUST present the weight data for Fig 4E, 7B and G as well as 8A as absolute values as well. I cannot understand how a group variability could impact the conclusion if there is a 60% or higher body weight gain. These Figures can be presented in the supplemental material but they should be accessible for the reader to assess the data.

We modified all the weight data to absolute values as requested.

Corresponding Author Name: Andrea Ballabio

Journal Submitted to: Embo Molecular Medicine

Manuscript Number: EMM-2016-07204-V2-Q